# TOWARDS DISTRIBUTION-AWARE ACTIVE LEARNING FOR DATA-EFFICIENT NEURAL ARCHITECTURE PREDICTOR

## ABSTRACT

As the neural predictor (NP) provides a fast evaluation for neural architectures, it is highly sought after in neural architecture search (NAS). However, the high computational cost involved in generating training data results in its scarcity, which in turn limits the accuracy of the NP. Active learning (AL) has the potential to address this issue by prioritizing the most informative samples, yet existing methods struggle with selection bias when faced with imbalanced data distributions, often prioritizing diversity over representativeness. In this paper, we redefine the sample selection mechanism in AL and propose a Distribution-aware Active Learning framework for Neural Predictor (called **DARE**). The goal is to select samples that not only ensure diversity but also exhibit a high degree of generalizability, making them more representative of the underlying data distribution. Our approach first extracts architecture representations via a graph-based encoder enhanced with a consistency-driven objective. Then, a two-stage selection strategy identifies both globally diverse and locally reliable samples through progressive representation learning and refinement. For non-uniform data distributions, we further introduce an adaptive mechanism that anchors sampling to key regions with high similarity density, avoiding performance degradation caused by outliers. Extensive experiments have shown that the proposed distribution-aware active learning strategy samples a higher-quality training dataset for NPs, allowing the neural architecture predictor to achieve state-of-the-art results.

## 1 INTRODUCTION

In recent years, neural architecture search (NAS) (Elsken et al., 2019; Song et al., 2024; Yu et al., 2024; Salmani Pour Avval et al., 2025) has gained increasing attention as a powerful technique for automatically discovering optimal neural architectures. NAS has shown great promise in various domains, including but not limited to computer vision (CV) (Gao et al., 2023; Poyser & Breckon, 2024), machine learning (ML) (Salehin et al., 2024), and natural language processing (NLP) (Chen et al., 2024). However, a decent search capacity of traditional NAS often comes with a high cost in terms of time or computational resources. Therefore, it is urgent to design effective and reasonable acceleration strategies. Low-fidelity training, a common acceleration strategy in NAS, reduce evaluation time by shortening training epochs, dataset size, *etc.*, but may lead to inaccurate performance predictions and the omission of superior architectures due to insufficient training (Liu et al., 2022), as highlighted by Zhou (Zhou et al., 2020).

An alternative to accelerate the process of NAS is leveraging a neural predictor (NP) to estimate the performance of neural architectures, obviating the high cost of model training in evaluation. Due to the superior characteristics of the NP, it soon won the attention of researchers. However, achieving precise evaluations from the NP requires substantial training samples, each involving training and testing over several hours or days (Liu et al., 2022; Deng et al., 2017). Therefore, considering the limited feasibility of acquiring a large number of labeled samples, there is an urgent need to extract the most informative samples from the existing data under resource constraints, thereby enhancing model performance. This raises the first major challenge: **I)** ***How to effectively select highly informative architectures for training the predictor under a severely limited labeling budget?***

Given the pivotal role of training samples, Active Learning (AL) (Li et al., 2024a) techniques provide a new paradigm to address this challenge as an efficient optimization strategy. AL intelligently selects the most informative samples, achieving the greatest performance improvement with minimal labeling cost. However, current AL methods (Li et al., 2024b; Lin et al., 2024) primarily focus on enhancing sample diversity, often overlooking the impact of data distribution characteristics on sample selection. As a result, the selected samples lack representativeness and fail to accurately reflect the distribution of the dataset, particularly evident in neural architecture datasets. This leads to the second key challenge: **II)** *How to balance diversity and distributional representativeness in the sampling process to ensure more effective predictor training?*

To solve both challenges, we redefine the sample selection mechanism in AL and propose a Distribution-aware Active Learning framework for Neural Predictor (called **DARE**). The goal is to efficiently sample instances that exhibit diversity and a high degree of generalization while accounting for dataset-specific distributions. Specifically, in each active learning iteration, we first extract architecture embeddings using a graph-based encoder, trained with a consistency-preserving objective to improve representation quality. Based on these embeddings, we perform a two-stage sample selection process. The first stage identifies globally diverse candidates by computing pairwise distances between labeled and unlabeled architectures. The second stage enhances local reliability by leveraging proximity topology, such as clustering and Delaunay Triangulation calculation, to refine the neighborhood around labeled samples and assign pseudo-labels for retraining. Moreover, to address the common issue of non-uniform sample distributions in architecture spaces, we introduce an adaptive sampling mechanism that identifies anchor samples with strong coverage of the unlabeled pool and restricts sampling to informative intermediate-density regions. This ensures that the selected samples not only broaden the search space exploration but also align closely with the true data distribution, enabling the predictor to generalize more effectively.

We evaluate our proposed **DARE** on three widely used NAS search spaces, *i.e.*, NAS-Bench-101 (Ying et al., 2019), NAS-Bench-201 (Dong & Yang, 2020), and DARTS (Liu et al., 2018). The experimental results show that the NP achieves state-of-the-art prediction performance after training on samples selected by **DARE**. Furthermore, the **DARE** also significantly improves the NAS performance in the search for the optimal neural architecture. Additionally, we validate the effectiveness of **DARE** on the TransNAS-Bench-101 (Duan et al., 2021) across various tasks, where it also achieves impressive performance. Finally, we also perform an in-depth analysis to verify the superiority of the proposed strategy. In summary, our contributions are:

❶ *Problem Connection.* Paying attention to the significance of training samples, we establish a novel connection between neural predictors and sampling bias, emphasizing that the quality and representativeness of training data are critical for reliable architecture performance estimation. This is the first work that focuses on the training data of neural predictors.

❷ *Novel Framework.* We introduce **DARE**, a two-stage sampling framework that first selects diverse candidates via max-min strategy, then adaptively refines the sampling regions using a key-point-guided mechanism. This framework ensures both diversity and representativeness by explicitly connecting sampling behavior with data distribution.

❸ *Comprehensive Validation.* The experimental results show that **DARE** achieves state-of-the-art prediction performances under limited training data. In addition, we validate our method across different tasks and consistently achieve excellent performance.

## 2 RELATED WROK

### 2.1 NEURAL ARCHITECTURE PERFORMANCE PREDICTORS

Research on neural architecture performance predictors (NPs) has gained increasing attention, with existing approaches broadly categorized into learning curve-based (Ding et al., 2025) and model-based methods (Zhao et al., 2025). The former extrapolates final performance from partial training curves but suffers from instability and sensitivity to hyperparameters, often requiring multi-fidelity techniques (Falkner et al., 2018). Model-based methods are more widely adopted, where a regression model is trained directly on architecture representations. These include traditional machine learning methods (Sun et al., 2019; Luo et al., 2020), graph-based predictors (Mills et al., 2023; Shi et al.,

2020; Ning et al., 2022), and Bayesian frameworks (White et al., 2021). Recently, the high cost of labeled training data has motivated studies on improving sample efficiency through semi-supervised learning (Tang et al., 2020) and data augmentation (Liu et al., 2021). However, these methods rely on raw labeled samples, and the presence of low-quality initial samples can significantly compromise their effectiveness, ultimately leading to suboptimal outcomes. Our work proposes leveraging active learning strategies to construct high-quality training samples.

## 2.2 ACTIVE LEARNING

Active learning (AL) aims to select the most informative data points for labeling to optimize model performance with minimal labeled data. The acquisition strategies of AL can be simply divided into two categories: uncertainty-based methods and diversity-based methods. The uncertainty-based methods typically choose the sample with the lowest output probability of the model prediction for labelling (Fuchsgruber et al., 2024; Peng et al., 2021; Ji et al., 2023). The diversity-based methods are used to select the unlabeled samples that are least similar to the labelled samples to improve the performance of the classifier, which is the method used in this study. In addition, the diversity-based methods are commonly combined with clustering algorithms (Li et al., 2023) to achieve similarity comparisons between samples (Tan et al., 2024). While these methods excel at selecting diverse samples, they overlook the sampling bias introduced by data distribution. Our study proposes an adaptive sampling strategy that ensures diversity while providing high-quality training data for NP.

## 3 APPROACH

### 3.1 PROBLEM SETTING

NAS is to search from the space of neural architectures and identify the optimal one. Due to the high cost of the performance evaluation process in NAS, it is inevitable to design an efficient architecture evaluation method. To this end, we focus on accelerating this process with neural predictors.

Let the search space of the neural architectures be $\mathcal{X}$. Due to the limited budget, only parts of $\mathcal{X}$ are evaluated and obtain the ground truth $\mathcal{Y}$ (*i.e.*, the performance). Considering that there are tens of thousands of neural architectures in $\mathcal{X}$, it is not realistic to assign a true performance label to each network architecture, so the amount of data in $\mathcal{Y}$ is far less than $\mathcal{X}$, that is, $\mathcal{D} = \{\mathcal{X}; \mathcal{Y}\} = \{x_1, x_2, \ldots, x_p; y_1, y_2, \ldots, y_k\}$ ($k \ll p$). Then, according to the performance in $\mathcal{Y}$, the neural architectures with the labels are obtained from $\mathcal{X}$, forming the training data $\mathcal{D}_{train} = \{\mathcal{X}^l; \mathcal{Y}\} = \{x_1^l, x_2^l, \ldots, x_k^l; y_1, y_2, \ldots, y_k\}$. The performance predictor $P$ (a regression model) is trained with input $\mathcal{X}^l$, and the resulting output is compared with $\mathcal{Y}$. The objective function $J(\cdot)$ of this process can be formulated as:

$$J(\mathcal{W}, \mathscr{D}) = \frac{1}{|\mathscr{D}|} \sum_{i=1}^{|\mathscr{D}|} \mathcal{L}(P(\mathcal{W}, x_i), y_i), \tag{1}$$

where $\mathcal{W}$ is the training parameters of the regression model, $\mathscr{D}$ denotes the data involved in training, in this case $\mathcal{D}_{train}$, and $\mathcal{L}(\cdot)$ denotes the loss function of $P$. To get a well-performing predictor, it is necessary to put forward a high requirement for $\mathcal{X}^l$. Under the condition of keeping the number of samples unchanged, improving the quality of the samples is an effective way.

### 3.2 OVERVIEW

In this paper, we consider that the budget for annotating the samples is limited. We aim to effectively annotate the samples and thus obtain an accurate predictor for the final NAS. Formally speaking, let the budget of the annotation number be $K$. The problem of this paper is how to select these $K$ samples from the search space, such that the performance predictor can be well-trained. To address the challenge, we propose a distribution-aware active learning framework to select the most informative $K$ samples such that we can train an accurate NP based on them. Specifically, in each iteration of AL, we propose a two-stage max-min sampling strategy to ensure sample diversity (as illustrated in Figure 1). In the first stage, the predictor extracts embeddings and computes distances between labeled and unlabeled samples, selecting those with the largest and smallest distances. In the second stage, spatial partitioning techniques such as clustering and Delaunay triangulation are employed

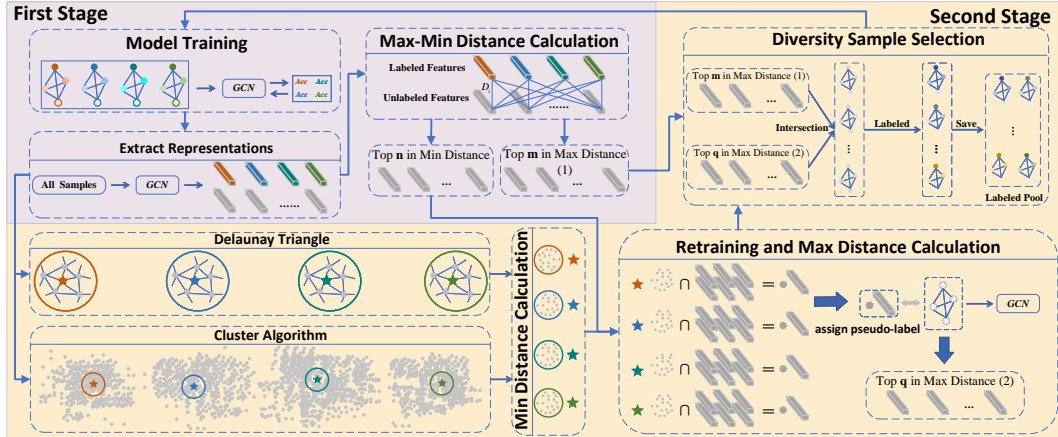

Figure 1: The flowchart of the Two-stage Max-Min Sampling. First stage (purple background): (1) Model Training. (2) Max-Min Distance Calculation. The distance between labeled and unlabeled samples is calculated using the cosine distance. Second stage (yellow background): (1) Min Distance Calculation. The unlabeled samples with the minimum distance from the labeled samples are obtained by Delaunay triangulation or a clustering algorithm. (2) Retraining and Max Distance Calculation. The minimum distance between unlabeled samples obtained from the two acquisitions is intersected and assigned pseudo-labels provided to the model for retraining. (3) Diversity Sample Selection.

to re-evaluate these distances and improve the reliability of the selected samples. To further handle the challenges introduced by non-uniform sample distributions, we incorporate a key-point guided adaptive sampling strategy that dynamically adjusts the sampling region, refining the candidate pool for more effective selection. The sampling framework ensures that the queried samples are both diverse and well-aligned with the underlying distribution.

### 3.3 TWO-STAGE MAX-MIN SAMPLING

Calculating the distance between samples to enhance diversity is a commonly effective method. However, when an accurate representation of the samples cannot be ensured, distance calculations may not capture sample diversity. Therefore, in this section, a two-stage distance calculation is proposed to ensure diversity. In the first stage, we refined the model training process to extract more accurate sample representations, which were then used for distance calculation. In the second stage, a semi-supervised mechanism was employed to further train the model and recompute the distances, enhancing the reliability of the results.

**First Stage: Model Training.** The neural architecture is a model with extremely tight internal connections, and both the order of the structures and the way of connection are key factors affecting the performance, so representing it as a graph structure can highlight this characteristic. Consequently, a model capturing such a graph structure is essential for processing graph-structured data. In this context, employing graph convolutional networks (GCNs) enhances the extraction of meaningful representations $\mathbb{R}$ from neural architecture graphs. Specifically, we adopted the GCN to deal with the bi-directional information flow of the neural architecture (Wen et al., 2020). The layers (or blocks) of the neural architecture are represented by nodes $I$ and transformed into a one-hot vector to form the operation matrix $O$. In addition, the adjacency matrix $A \in \mathbb{R}^{I \times I}$ denotes the relationship among nodes. Therefore, the representations $V$ of the sample at $ith$ layer can be expressed as follows:

$$\begin{cases} V_2 = \frac{1}{2} \text{ReLU}\left(AOW_1^+\right) + \frac{1}{2} \text{ReLU}\left(A^T OW_1^-\right) \\ V_i = \frac{1}{2} \text{ReLU}\left(AV_{i-1}W_{i-1}^+\right) + \frac{1}{2} \text{ReLU}\left(A^T V_{i-1}W_{i-1}^-\right), \end{cases} \tag{2}$$

where $W$ denotes the training parameters and $\text{RELU}(\cdot)$ denotes the activation function.

However, it is tough for the GCN to train with a small amount of data and obtain an accurate representation of the neural architecture, so we designed a new loss function to assist in training. Data augmentation (Liu et al., 2021; Ma et al., 2025) is an effective technique to increase the amount

of data, especially in image processing. In this study, we perform data augmentation for each neural architecture and compare the similarity between the augmented and original architectures (detailed in Appendix C). As a result, we convert the loss function from the original Equation 1 to:

$$\min_{\mathcal{W}} \frac{1}{k} \sum_{i=1}^{k} ((1 - \lambda) \times \mathcal{L}(P(\mathcal{W}, x_i^l), y_i) + \lambda \times \mathcal{S}(x_{i,ori}^l, x_{i,aug}^l)), \tag{3}$$

where $x_{i,ori}^l$ and $x_{i,aug}^l$ denote the representation of the $ith$ original and augmented sample, $\lambda$ is the equilibrium coefficient, and $k$ is the number of samples. In addition, $\mathcal{S}$ is the cosine distance.

**First Stage: Max-Min Distance Calculation.** Leveraging the trained model, we extracted the embeddings of all samples, including labeled and unlabeled ones, and computed the cosine distance $D_i$ ($i$ indicates the $ith$ labelled data) between each labeled sample and all unlabeled samples. Then, $\mathcal{X}_i^{max} \in \mathcal{X}^u$ ($|\mathcal{X}_i^{max}| = m$) and $\mathcal{X}_i^{min} \in \mathcal{X}^u$ ($|\mathcal{X}_i^{min}| = n$) are extracted via $D_i$, denoting the $m$ unlabeled samples furthest from the $ith$ labeled sample and the $n$ unlabeled samples nearest to $ith$ the labeled sample, respectively. Crucially, the current computation method is better suited for uniformly distributed datasets. However, when dealing with non-uniformly distributed datasets (as illustrated in Figure 2), the current maximum distance computation is not applicable (*c.f.,* Sec. 3.4).

**Second Stage: Min Distance Calculation.** For the initially obtained sets $\mathcal{X}_i^{max}$ and $\mathcal{X}_i^{min}$, their reliability and representativeness cannot be fully guaranteed due to potential noise and embedding inaccuracy. Therefore, a second refinement step is required to enhance selection precision, starting with a recalculation of the minimum-distance samples.

To improve the accuracy of nearest-neighbor selection, we further incorporate spatial partitioning methods to redefine sample neighborhoods from complementary perspectives: **I) Delaunay Triangulation**. We perform Delaunay triangulation of the region near the labelled samples, and an unlabeled sample node is considered neighbours (nearest samples) if it lies on the same side of at least one triangle as the nodes of the labelled samples. **II) Cluster Algorithm**. The clustering algorithm is used to divide the unlabeled samples into $L$ classes ($L$ denotes the number of labelled samples) and make the labelled samples the centre of the clusters to find neighbouring samples.

Based on the either method, $\mathcal{X}_i^{min^*} \in \mathcal{X}^u (|\mathcal{X}_i^{min^*}| = n^*)$ is obtained to denote the $n^*$ nearest unlabeled samples to the $ith$ labeled sample. Notably, the two methods are not used simultaneously. We prioritise Delaunay triangulation, as it captures the geometric structure of the sample space and yields more spatially coherent neighbors, making it well-suited for reliable distance refinement. However, due to its limited neighbor count, we also employ clustering to expand the neighborhood set and ensure sufficient overlap for subsequent intersection operations.

**Second Stage: Retraining and Max Distance Calculation.** Based on the nearest unlabeled samples obtained from the different rules above, we will first perform an intersection operation. Combining $\mathcal{X}_i^{min}$ and $\mathcal{X}_i^{min^*}$, the samples coexisting in the two sets are acquired, *i.e.*, $\mathcal{X}_{i,comb}^{min} = \mathcal{X}_i^{min} \cap \mathcal{X}_i^{min^*}$. Next, we assign pseudo-labels $\widetilde{\mathcal{Y}}$ to the $\mathcal{X}_{i,comb}^{min}$ and get the training data $D_{train}^* = \left\{ \mathcal{X}_{1,comb}^{min}, \ldots, \mathcal{X}_{k,comb}^{min}; \widetilde{\mathcal{Y}} \right\}$ ($k$ is the number of labelled samples in the labelled pool currently available). Finally, we use $D_{train}^*$ to re-train the NP.

After re-training, the NP will extract the representations of all samples once again. Based on these new representations, we perform the maximum distance calculation and obtain $\mathcal{X}_i^{max^*} \in \mathcal{X}^u$ ($|\mathcal{X}_i^{max^*}| = m^*$), *i.e.*, the furthest $m^*$ unlabeled samples from the $ith$ labelled sample (note that the difference of $\mathcal{X}_i^{max}$ and $\mathcal{X}_i^{max^*}$ is that the unlabeled sample representations are different when participating in the distance calculation).

**Second Stage: Diverse Sample Selection.** Diversity selection aims to identify the samples with the lowest similarity between the obtained unlabelled samples and the existing labelled samples. So, we perform the following intersection of unlabeled samples:

$$\begin{cases} \mathcal{X}_{i,comb}^{max} = \left\{ \mathcal{X}_i^{max} \cap \mathcal{X}_i^{max^*}; i = 1, \ldots, K \right\} \\ \mathcal{X}_{final} = Top(\mathcal{X}_{i,comb}^{max}, n_s), \end{cases} \tag{4}$$

where the resulting $\mathcal{X}_{final}$ is the final selected unlabeled samples, which are also the least similar to the labelled sample. In addition, $n_s$ denotes the number of samples selected in each iteration, and

$Top(\cdot)$ denotes the $n_s$ samples selected from $\mathcal{X}_{i,comb}^{max}$ with the greatest distance. Note that $m^*$ is less than $m$ to reduce the impact of pseudo-labeled samples.

### 3.4 KEY-POINT GUIDED ADAPTIVE SAMPLING

As detailed above, for $\mathcal{X}_i^{max}$, we have to consider the scenario where the sample similarity is non-uniformly distributed. In Figure 2, for example, we plot the non-uniform distribution of the sample in the search space based on similarity. It can be seen that the samples are mainly concentrated in the green circles, while those outside the green circle are not only sparse but also very dispersed. Experimentally, it was found that if more attention was paid to the sample outside the green circle, which was considered to be more diverse, the performance of the trained predictor was poor.

Therefore, we design a novel Key-point Guided Adaptive sampling method for the scenario of non-uniform distribution of sample similarity. The details are shown in the equations below:

$$\begin{cases} x_{key} = \operatorname{argmin}_{x_i^l \in \mathcal{X}^l} \left( \frac{1}{T} \sum_{t=1}^T U(x_i^l) \right) \\ U(x_i^l) = \sum_{x_j^u \in \mathcal{X}^u} \mathcal{S}\left(x_i^l, x_j^u\right), j = 1 \ldots z, \end{cases} \tag{5}$$

where $x_{key}$ (*i.e.*, key point) is the labelled sample with the minimum average distance from $z$ randomly selected unlabeled samples, $T$ denotes the number of repetitions, $\mathcal{S}(\cdot)$ represents the cosine distance, and $\mathcal{X}^u$ denotes the unlabeled data set. We assume that the resulting $x_{key}$ has the highest proportion of similarity to unlabeled samples, which can cover more information about unlabeled samples. Then, we will select samples with the maximum distance between intervals $[|D_{x_{key}}| \times \alpha, |D_{x_{key}}| \times \beta]$ (the interval corresponds to the blue region in the Figure 2), where $D_{x_{key}}$ is the distance from $x_{key}$ to all unlabeled samples (sorted from smallest to largest), $\alpha$ and $\beta$ are coefficients and $\alpha < \beta$. Note that in a uniformly distributed scenario, the maximum distance calculation is performed for each labelled sample, whereas in a non-uniformly distributed scenario, the maximum distance is only calculated for key samples.

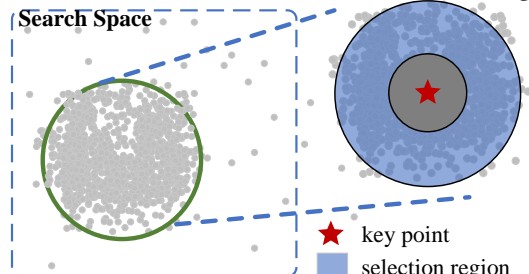

Figure 2: The calculation of the maximum distance between unlabelled and labelled samples is performed in scenarios with a non-uniform distribution of similarity.

## 4 THEORETICAL ANALYSIS

To achieve superior performance of the NP, we hope that the selected samples remain as diverse as possible while selecting as few samples as possible. On the other hand, we have found experimentally that the NP performs better when the distribution of the selected sample approximates the total sample. Proceeding from this, we provide a brief analysis of the minimum number of selected samples when the distributions are similar. The details are shown in the Appendix D.

Let $\mathcal{D}$ and $\mathcal{D}_{train}$ be the dataset containing all samples and the dataset of the selected samples (used as training data). $\cos(\cdot)$ denotes the cosine similarity calculation between samples. The following proposition gives the minimum value of $n$ (*i.e.*, the number of selected samples) based on the cosine similarity between the samples.

**Proposition 4.1** *Let $\mathcal{D}_{train} = \{X_1, X_2, \ldots, X_n\}$, and $\bar{X} = \frac{1}{n} \sum_{i=1}^n X_i$. With probability of $1 - \delta$, we have:*

$$n \geq \sqrt{\frac{\sum_{i=1}^n (b_i - a_i)^2}{2t^2} \ln \frac{2}{\delta}}, \quad t = \frac{1}{n(n-1)} \sum_{i=0}^n \sum_{j \neq i}^n \left(1 - \cos(X_i, X_j)\right) + \epsilon, \tag{6}$$

*where $\epsilon$ is a very small positive number to avoid the situation that the denominator is 0 in Equation 6.*

The proposition gives the bounded value of $n$, which is subject to the variable $t$. This means that the smaller the cosine similarity among the samples, the smaller the value of $n$, *i.e.* the higher the sample diversity and the smaller the number of samples to be selected. Notably, this analysis relies on simplified static assumptions and aims to provide theoretical intuition for the sampling strategy.

Table 1: Comparison results of *DARE* with the SOTA methods on two datasets. "–" indicates the indicator could not be reproduced or has no available public report. Bold indicates the best result.

| Methods | NAS-Bench-101 | | | Methods | NAS-Bench-201 | | |
|---|---|---|---|---|---|---|---|
| | $K$ | KTau↑ | MSE↓ | | $K$ | KTau↑ | MSE↓ |
| Peephole | 1K | 0.4373±0.0112 | 0.0071±0.0005 | Peephole | 150 | 0.5112±0.0311 | 0.0062±0.0010 |
| E2Epp | 1K | 0.5705±0.0082 | 0.0042±0.0003 | | 78 | 0.4561±0.0280 | 0.0077±0.0009 |
| SSANA | 1K | 0.6541±0.0078 | 0.0031±0.0003 | E2Epp | 150 | 0.6699±0.0100 | 0.0013±0.0008 |
| HAAP | 1K | 0.7126±0.0024 | 0.0023±0.0003 | | 78 | 0.5729±0.0193 | 0.0019±0.0006 |
| HAAP | 424 | 0.7010±0.0022 | 0.0024±0.0003 | HAAP | 150 | 0.7375±0.0200 | **0.0005±0.0001** |
| RFGIAug | 424 | 0.6513±0.0026 | 0.0019±0.0002 | | 78 | 0.6619±0.0219 | 0.0011±0.0003 |
| ReNAS | 424 | 0.6619±0.0033 | 0.0021±0.0005 | RFGIAug | 150 | 0.7219±0.0019 | 0.0009±0.0002 |
| NPNAS | 424 | 0.6743±0.0029 | 0.0027±0.0003 | | 78 | 0.6941±0.0008 | 0.0010±0.0002 |
| MLP | 381 | 0.5116±0.0011 | 0.0058±0.0001 | ReNAS | 150 | 0.6731±0.0041 | 0.0008±0.0011 |
| LSTM | 381 | 0.5874±0.0017 | 0.0046±0.0002 | | 78 | 0.6210±0.0190 | 0.0012±0.0005 |
| BOGCN | 381 | 0.5790 | — | NPNAS | 150 | 0.7004±0.0031 | 0.0009±0.0002 |
| HOP-2 | 190 | 0.6440 | — | | 78 | 0.6635±0.0195 | 0.0010±0.0017 |
| *DARE* | 1K | **0.7576±0.0030** | **0.0014±0.0001** | *DARE* | 150 | **0.7854±0.0050** | **0.0005±0.0002** |
| | 424 | 0.7311±0.0031 | 0.0018±0.0002 | | | | |
| | 381 | 0.6814±0.0030 | 0.0018±0.0001 | | 78 | 0.7014±0.0300 | 0.0008±0.0003 |
| | 190 | 0.6593±0.0030 | 0.0020±0.0003 | | | | |

## 5 EXPERIMENTS

### 5.1 EXPERIMENTAL SETUP

**Datasets.** This study mainly carries out experiments on search spaces, *i.e.*, NAS-Bench-101 (NB101) (Ying et al., 2019), NAS-Bench-201 (NB201) (Dong & Yang, 2020), DARTS (Liu et al., 2018), and TransNAS-Bench-101 (TransBench-101) (Ying et al., 2019). More detail can be found in Appendix E.

**Baselines.** We compare *DARE* with multiple state-of-the-art methods. The competitors include: Peephole (Deng et al., 2017), E2Epp (Sun et al., 2019), SSANA (Tang et al., 2020), HAAP (Liu et al., 2021), RFGIAug (Xie et al., 2023), ReNAS (Xu et al., 2021), NPNAS (Wen et al., 2020), MLP (Wang et al., 2019), LSTM (Wang et al., 2019), BOGCN (Shi et al., 2020), HOP-2 (Chen et al., 2021b). In the architectural search for real scenarios, we also introduce additional comparison algorithms, including: GATES (Ning et al., 2022), NASBOT (White et al., 2020), ResNet (He et al., 2016), TNASP (Lu et al., 2021), PINAT (Lu et al., 2023), NAR-Former (Yi et al., 2023), BRP-NAS (Dudziak et al., 2020), MeCo (Jiang et al., 2024), SWAP (Peng et al., 2024), REA (Dong & Yang, 2020), RS (Dong & Yang, 2020), HNAS (Shu et al., 2022), and RoBoT (He et al., 2024).

**Implementation Details and Evolution Metrics.** We use the GCN model as a predictor, and the input to the predictor is the representation of the architecture in the form of multiple matrices, following the setting in (Liu et al., 2021). The selection in AL is divided into two parts, in which 5 samples are randomly selected for annotation at initialization, while 10 samples are subsequently selected at each iteration using the proposed sampling method. The predictor is trained using the Adam optimizer for 300 epochs with a learning rate of 0.001, and the batch size is the same as the number of samples selected at each iteration. The $\lambda$ in Eq. equation 3 is 0.3. We evaluate the predictor using Kendall's Tau (KTau), Mean Squared Error (MSE), and Rank to assess ranking consistency and regression accuracy, while the top-1 architecture's performance is further reported using validation (Val) and test (Test) accuracy. In addition, for the DARE sampling strategy, the hyperparameters are adjusted according to the search space scale. For NAS-Bench-101, the size of the farthest candidate set $m$ is set to 10,000, and the nearest candidate set $n$ is 10,000 in the first stage. The refined farthest candidate size $m^*$ is 10,000, while the key-point sampling parameters (Equation 5) are set to $z = 1,000$ and $T = 10$. For NAS-Bench-201 and DARTS, these parameters are set to $m = 1,000$, $n = 1,000$, $m^* = 1,000$, $z = 500$, and $T = 10$, respectively.

### 5.2 EMPIRICAL RESULTS

**Comparison Results.** We first compare the *DARE* with several SOTA methods on two datasets to verify its effectiveness. The comparison results are shown in Table 1. *DARE consistently outperforms*

Table 2: Search results of *DARE* with the SOTA methods on three datasets.

| NAS-Bench-101 | | | | NAS-Bench-201 | | | DARTS | | | |
|---|---|---|---|---|---|---|---|---|---|---|
| Methods | $K$ | Accuracy(%)↑ | Rank↓ | Methods | Val ACC.(%) | Test ACC.(%) | Methods | $K$ | CIFAR10 | ImageNet |
| Peephole | 1K | 93.41±0.34 | 1922 | NASBOT | - | 93.64±0.23 | TNASP | 1000 | 97.48 | 75.50 |
| E2Epp | 1K | 93.77±0.13 | 687 | E2Epp | 90.61±0.89 | 93.39±0.75 | PINAT | 1000 | 97.58 | 77.80 |
| SSANA | 1K | 94.01±0.12 | 59 | HAAP | 91.18±0.25 | 94.00±0.25 | NAR-For | 100 | 97.52 | - |
| ReNAS | 1K | 93.95±0.11 | 148 | NPNAS | 91.27±0.29 | 93.95±0.28 | BRP-NAS | 60 | 97.52 | - |
| HAAP | 1K | 94.09±0.11 | 16 | ReNAS | 90.90±0.31 | 93.99±0.25 | MeCo | - | 97.36 | - |
| BOGCN | 381 | — | 1362 | ResNet | 90.83 | 93.97 | SWAP | - | 97.61 | 76.00 |
| GATES | 381 | — | 22 | *optimal* | 91.61 | 94.37 | *DARE* | 100 | **97.63** | **78.01** |
| *DARE* | 381 | **94.11±0.11** | **6** | *DARE* | **91.47±0.14** | **94.06±0.30** | *DARE* | 60 | 97.55 | 77.13 |

*all baseline methods across both NB101 and NB201 datasets under various training sample sizes.* On NB101, *DARE* achieves the highest ranking accuracy with a KTau improvement of up to 6.31% over the best baseline (HAAP) when using 1K training samples, while reducing the regression error (MSE) by 39.13%. Even with fewer samples (*e.g.*, $K = 424$), it still outperforms strong baselines such as NPNAS by 8.42% in KTau and 33.33% in MSE, demonstrating its effectiveness in low-resource settings. On NAS-Bench-201, *DARE* achieves a KTau of 0.7854 with $K = 150$, surpassing the second-best method (HAAP) by 6.49%, while maintaining the lowest MSE of 0.0005. When the sample size drops to 78, it still maintains a 1.05% lead in KTau over the next best performer, indicating that *DARE* selects more informative and generalizable samples under strict budget constraints.

**Results of NAS** To test the performance of the *DARE* further, we apply the proposed algorithm to a real scenario to perform a search for the optimal network architecture. *The search results in Table 2 demonstrate the strong generalization performance of architectures discovered by DARE across three benchmark datasets.* On NB101, *DARE* achieves an accuracy of 94.11% with only 381 training samples, surpassing all other baselines including HAAP and ReNAS, and ranking 6-*th* among all 423k architectures, which represents a substantial improvement in sample efficiency. On NAS201, the architecture selected by *DARE* achieves 94.06% test accuracy, outperforming all baseline predictors and even approaching the performance of the optimal architecture (94.37%) identified through exhaustive evaluation. On the DARTS benchmark, *DARE* also achieves state-of-the-art performance with only 100 queried samples, reaching 97.63% on CIFAR-10 and 78.01% on ImageNet, outperforming existing NAS approaches such as TNASP, PINAT, and SWAP. These results collectively verify the effectiveness of *DARE* in identifying high-performing architectures under limited supervision across diverse search spaces and task types.

**Application on Various Tasks.** To validate the effectiveness of our sampling strategy, we further conduct experiments on various tasks from TransBench-101 (Micro). Additional results are provided in the Fig. 12. As shown in Table 3, *DARE* achieves the best performance in all four tasks, including 54.91% accuracy on scene classification and 95.01% on jigsaw puzzle recognition, both surpassing all compared methods. On object classification and semantic segmentation, *DARE* reaches 45.59% accuracy and 94.61% mIoU, matching the optimal values reported in the benchmark. Compared to representative baselines such as HNAS, *DARE* demonstrates consistent improvements across all tasks, highlighting its superior ability to select architectures that generalize well in multi-task.

Table 3: Search results on TransBench-101.

| Methods | Accuracy (%) | | | mIoU (%) |
|---|---|---|---|---|
| | Scene | Object | Jigsaw | Segment. |
| REA | 54.63 | 44.92 | 94.81 | 94.55 |
| RS | 54.56 | 44.76 | 94.63 | 94.53 |
| HNAS | 54.29 | 44.08 | 94.56 | 94.57 |
| RoBoT | 54.87 | 45.59 | 94.76 | 94.58 |
| DEAR | **54.91** | **45.59** | **95.01** | **94.61** |
| *Optimal* | 54.94 | 45.59 | 95.37 | 94.61 |

**Ablation Study.** We conduct ablation studies on NB101 to assess the contributions of each component in our framework. *DARE* w/o FS or SS denotes removing the first stage or second stage max-min sampling strategy, *DARE* w/o TMMS denotes removing the two-stage max-min sampling strategy, *DARE* w/o KGA indicates removing the key-point guided adaptive sampling strategy. Notably, KGA does not exist without TMMS, as it depends on TMMS. In addition, NB101 is a non-uniformly distributed dataset processed to obtain a uniform distri-

Table 4: Ablation Study on NB101.

| Distribution | Uni. | Non-Uni. |
|---|---|---|
| w/o FS | 0.6628 | 0.6518 |
| w/o SS | 0.6564 | 0.6834 |
| w/o TMMS | 0.6127 | 0.6255 |
| w/o KGA | 0.6943 | 0.7122 |
| *DARE* | 0.7019 | 0.7311 |

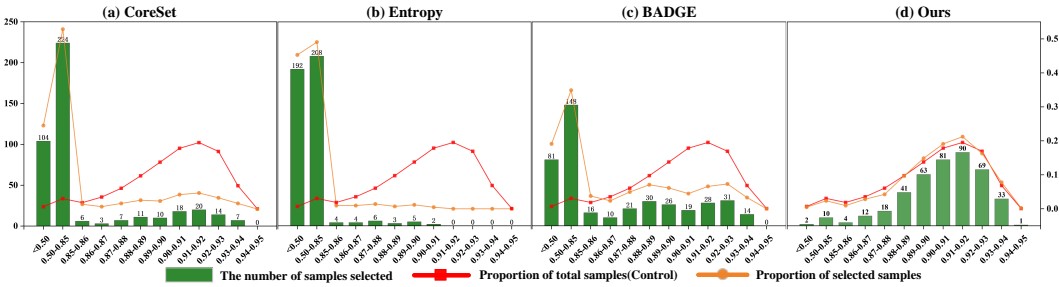

Figure 4: Distribution of samples selected by different AL methods. The x-axis coordinate indicates the accuracy range, the y-axis coordinate (left) indicates the number of samples selected by the sampling strategy in different accuracy ranges, and the y-axis coordinate (right) indicates the proportion of samples to the total samples.

bution to evaluate the behavior of the sampling strategies under different conditions. As shown in Table 4, either w/o FS or w/o SS will reduce performance to some extent. While removing TMMS leads to a significant performance drop in both settings, this indicates its importance for ensuring sample diversity and representativeness. In contrast, the impact of KGA is more pronounced under non-uniform distributions, where it adapts the sampling region to account for data imbalance.

**Sensitive Analysis.** The sensitivity analysis illustrates the impact of the sampling range (*i.e.,* $\alpha$ and $\beta$) in Key-point Guided Adaptive Sampling. As shown in the Figure 3, the choice of sampling range has a significant impact on model performance. When the selected range is centered (*i.e.*, 0.4–0.5), the model achieves the highest KTau on both datasets, indicating that samples in this region are more representative. In contrast, when the range is too narrow and close to the key point (*i.e*., 0.0–0.1), the selected samples tend to be redundant and offer limited additional information. On the other hand, selecting from a range too far from the key point (*i.e.*, 0.9–1.0) leads to performance degradation as these samples may lie in sparse or noisy regions of the space, reducing the reliability of the predictor. These findings highlight the importance of carefully setting the sampling interval to balance informativeness and stability.

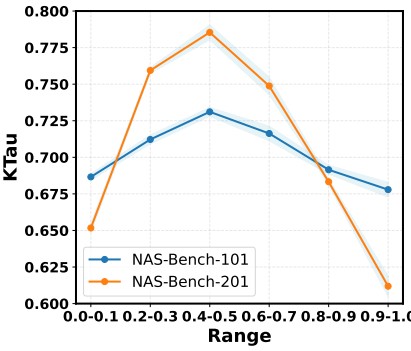

Figure 3: Sensitive analysis of $\alpha$ and $\beta$.

**Expansion Experiments on different AL.** To comprehensively evaluate the effectiveness of our active learning strategy, we conducted a fair comparison against several state-of-the-art methods, including NoiseStability (Li et al., 2024b), EOAL (Safaei et al., 2024), and BAL (Li et al., 2023), on the NAS-Bench-101 dataset. In the experiment, neural predictors were trained using 424 samples, with KTau and MSE as the evaluation metrics. As shown in Table 5, the results strongly demonstrate the superiority and robustness of our method. It achieved a KTau of 0.7311, significantly surpassing all baselines and

Table 5: Comparison of AL Strategies on NB101

| Methods | KTau | MSE |
|---|---|---|
| NoiseStability | 0.5817±0.0055 | 0.0024±0.0005 |
| EOAL | 0.6472±0.0035 | 0.0020±0.0003 |
| BAL | 0.6351±0.0072 | **0.0014±0.0003** |
| *DARE* | **0.7311±0.0031** | 0.0018±0.0002 |

showcasing a superior ability to rank architectural performance. Meanwhile, although its MSE (0.0018) is slightly higher than that of BAL (0.0014), it remains highly competitive. Moreover, its substantial advantage in KTau confirms that its overall performance is optimal for tasks that rely on precise ranking, such as neural architecture search.

**In-depth Analyze** In this part, we conduct an in-depth analysis of the samples obtained through different AL methods, including CoreSet (Sener & Savarese, 2018), Entropy (Wang & Shang, 2014), and BADGE (Ash et al., 2020). As discussed earlier, our goal is to ensure that the selected samples excel in both diversity and representativeness. In Fig 4, the histogram is the number of samples selected. The red line refers to the proportion of the number of samples in the corresponding accuracy range to the total number of samples in the NB101. The orange line refers to the proportion of the number of samples selected by AL methods in the corresponding accuracy range to the total number

of samples selected (the total number is 424). The comparison with three other AL methods reveals that the samples selected by our strategy closely align with the original data distribution. Compared to the baselines, our method avoids over-concentrating on low-accuracy regions and instead selects more samples from moderate- and high-accuracy intervals. This balanced distribution not only reflects the underlying data characteristics more faithfully but also helps improve the generalization ability of the predictor by covering a wider performance spectrum.

**More Experiments.** For additional experimental results and in-depth analysis, please refer to Appendix F-I. Furthermore, in Appendix I, we conducted a neural architecture search on the Transformer-based structure, which also obtained effective results.

## 6 CONCLUSION

This study highlights the critical role of training samples in the training process of neural predictors and proposes a novel and significant distribution-aware active learning method tailored for neural architecture datasets. The method incorporates a two-stage max-min selection strategy to ensure the diversity of selected samples and introduces a key-point-guided adaptive sampling strategy to enhance their representativeness, thereby comprehensively improving sample quality. Extensive experiments across various datasets validate the effectiveness and advantages of the proposed approach.

## ETHICS STATEMENT

All authors have read and adhered to the ICLR Code of Ethics. This research is foundational in nature, focusing on improving the computational efficiency of Neural Architecture Search (NAS) through a novel active learning strategy. Our work exclusively utilizes publicly available benchmark datasets (e.g., NAS-Bench-101, NAS-Bench-201, DARTS), which do not contain personally identifiable information or sensitive data, and no human subjects were involved in this study. Furthermore, our work does not present any other ethical violations.

## REPRODUCIBILITY STATEMENT

To ensure the reproducibility of our work, we have provided comprehensive details throughout the paper and its appendix. All experiments are conducted on publicly available and cited benchmark datasets, including NAS-Bench-101, NAS-Bench-201, DARTS, and TransNAS-Bench-101, as detailed in Sec. 5.1. Our proposed **DARE** framework and its components are thoroughly described in Sect.3. Specific implementation details, including hyperparameters, model configurations, and training procedures, can also be found in Sec. 5.1 and further elaborated in Appendix C. The theoretical analysis presented in Sect. 4 is supported by a complete proof in Appendix D. The source code for our experiments will be made publicly available upon the paper's acceptance to facilitate further research and verification.

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

## A    STATEMENT ON THE USE OF LARGE LANGUAGE MODELS

During the preparation of this manuscript, we used the Large Language Model (LLM) to polish the language and correct grammatical errors to improve readability. The LLM was not involved in any core research aspects of the paper, such as research ideation, experimental design, or analysis of results.

## B    OVERVIEW

The pipeline of the ***DARE*** is shown in Algorithm 1. Note that several randomly selected labelled samples are used for the first training of the NP before the AL strategy is applied.

As can be seen in the algorithm, the GCN is first trained on randomly selected labelled samples, after which the trained GCN is taken into the loop. In each iteration, the trained GCN performs representation extraction on all samples ($\mathbb{R}^l$, $\mathbb{R}^u$). Based on the representations, we perform the first round of computation of the maximum and minimum distance between labeled and unlabeled samples. Immediately after that, a second round of computation is performed, first to re-obtain the sample set of minimum distances by proximity topology. The two nearest sample sets are intersected and assigned pseudo-labels. The GCN is retrained on the data with pseudo-labels. Finally, the trained GCN is used to extract the representation of the sample again ($\mathbb{R}^{l*}$, $\mathbb{R}^{u*}$), and the sample with the farthest distance is selected according to the calculated distance. Intersection is done on the two most distant sample sets to obtain a diversity sample. After several iterations or satisfying the termination conditions, the final GCN is output.

---

**Algorithm 1: *DARE* algorithm**

---

**Input:** GCN: neural predictor; $K$: the total number of labeled samples; $n_s$: the number of samples selected in each iteration; $\mathcal{X}$: all samples; $\mathcal{X}^l$: labeled pool; $k$: the size of $\mathcal{X}^l$;

**Output:** GCN: trained predictor

1  $D_{train} = \left\{ \mathcal{X}^l, \mathcal{Y} \right\}$

2  GCN $\leftarrow Train(\text{GCN}, D_{train})$                                                  ▷ model training

3  **while** $k < K$ **do**

4  $\quad$ $\mathbb{R}^l, \mathbb{R}^u \leftarrow \text{GCN}(\mathcal{X})$                                          ▷ get representations

5  $\quad$ $\mathcal{X}^{max}, \mathcal{X}^{min} \leftarrow Dist(\mathbb{R}^l, \mathbb{R}^u)$                          ▷ calculate distance

6  $\quad$ $\mathcal{X}^{min^*} \leftarrow Select(\mathcal{X}^u_f)$                                      ▷ select nearest samples

7  $\quad$ $\mathcal{X}^{min}_{comb} \leftarrow \mathcal{X}^{min} \cap \mathcal{X}^{min^*}$

8  $\quad$ $D^*_{train} \leftarrow \left\{ \mathcal{X}^{min}_{comb}, \widetilde{y} \right\}$                            ▷ assign pseudo-labels

9  $\quad$ GCN $\leftarrow Train(\text{GCN}, D^*_{train})$

10  $\quad$ $\mathbb{R}^{l*}, \mathbb{R}^{u*} \leftarrow \text{GCN}(\mathcal{X})$

11  $\quad$ $\mathcal{X}^{max^*} \leftarrow Dist(\mathbb{R}^{l*}, \mathbb{R}^{u*})$

12  $\quad$ $\mathcal{X}_{final} \leftarrow \mathcal{X}^{max} \cap \mathcal{X}^{max^*}$                            ▷ select diverse samples

13  $\quad$ $\mathcal{X}^l \leftarrow \mathcal{X}^l \cap Top(\mathcal{X}_{final}, n_s)$

14  $\quad$ GCN $\leftarrow Train(\text{GCN}, \mathcal{X}^l)$

15  **end**

16  **Return:** GCN

---

## C    TECHNICAL DETAILS

**Data Augmentation.** In Sec. 3.3 (First Stage), we introduce a data augmentation mechanism to enable the GCN to learn accurate topological representations of architectures. Specifically, each architecture is represented as a computational graph, encoded by a node-type matrix and an adjacency matrix, where nodes signify operations and edges denote connections. To augment the data, we fix the input and output nodes while permuting the order of the remaining intermediate nodes. When the node-type matrix is reordered, the rows and columns of the adjacency matrix are rearranged synchronously. This process ensures that the network's topology and semantics remain strictly

identical. By exposing the GCN to multiple isomorphic representations of the same architecture, this method compels it to learn intrinsic structural features that are invariant to the node ordering.

**Architectural Representation.** Each neural architecture is formally represented as a Directed Acyclic Graph (DAG), where nodes correspond to specific network operations (*e.g.*, convolution) and directed edges represent the flow of data between them, with special nodes designated for the overall input and output. To make this graphical structure processable by a Graph Convolutional Network (GCN), we encode it into a matrix format. Specifically, each node's operation is encoded as a one-hot vector, and these are collectively stacked to form a node-type matrix. Concurrently, the graph's connectivity is captured by an adjacency matrix, where an entry of 1 signifies a connection and 0 signifies its absence. Consequently, each architecture is uniquely described by this pair of matrices, the node-type matrix and the adjacency matrix, which together serve as the input to the GCN.

## D  THEORETICAL ANALYSIS

To achieve superior performance of the NP, we hope that the selected samples remain as diverse as possible while selecting as few samples as possible. On the other hand, we have found experimentally (**In-depth Analysis**) that the NP performs better when the distribution of the selected sample approximates the total sample. Proceeding from this, we theoretically analyse the minimum number of selected samples when the distributions are similar.

We first introduce Lemma 1 (Hoeffding's inequality (Hoeffding, 1994)), which is a theorem in probability theory, and further, we deduce the minimum number of selected samples to be taken.

**Lemma 1.** (Hoeffding, 1994) *Let $X_1, X_2, \ldots, X_n$ be a collection of $n$ independent random variables, each with support in the intervals $[a_i, b_i]$, and let the expected value be $\mu = \frac{1}{n} \sum_{i=1}^{n} E[X_i]$. For any $t > 0$, it holds that:*

$$P\Big(\Big|\frac{1}{n}\sum_{i=1}^{n} X_i - \mu\Big| \geq t\Big) \leq 2\exp\Big(\frac{-2n^2 t^2}{\sum_{i=1}^{n}(b_i - a_i)^2}\Big). \tag{7}$$

**Proposition 1.** *Let $\mathcal{D}_{train} = \{X_1, X_2, \ldots, X_n\}$, and $\bar{X} = \frac{1}{n}\sum_{i=1}^{n} X_i$. With probability of $1 - \delta$, we have:*

$$n \geq \sqrt{\frac{\sum_{i=1}^{n}(b_i - a_i)^2}{2t^2} \ln \frac{2}{\delta}}, \tag{8}$$

and

$$t = \frac{1}{n(n-1)}\sum_{i=0}^{n}\sum_{j \neq i}^{n} \left(1 - \cos\left(X_i, X_j\right)\right) + \epsilon. \tag{9}$$

Let $\mathcal{D}$ and $\mathcal{D}_{train}$ be the dataset containing all samples and the dataset of the selected samples (used as training data). We can translate the bias calculations for the variables in the Lemma into differences in sample distributions. For $n$ samples $X_1, X_2, \ldots, X_n$, the sample mean is $\bar{X} = \frac{1}{n}\sum_{i=1}^{n} X_i$, and the sample variance is $S^2 = \frac{1}{n-1}\sum_{i=1}^{n}(X_i - \bar{X})^2$. Then Hoeffding's inequality can be written in the following form:

$$P\Big(\Big|\bar{X} - \mu\Big| \geq t\Big) \leq 2\exp\Big(\frac{-2n^2 t^2}{\sum_{i=1}^{n}(b_i - a_i)^2}\Big), \tag{10}$$

where $\mu$ represents the excepted value in $\mathcal{D}$, and $\sum_{i=1}^{n}(b_i - a_i)^2$ is the upper and lower bound of all sample values.

We can solve the inequality about $n$ by restricting the value of the upper bound on the probability of the right-hand side of the inequality. Specifically, assuming we want the upper bound of the probability to be less than probability $\delta$, that is:

Table 6: More Comparison results of **DARE** with the SOTA algorithms on NAS-Bench-101.

| Method | 1K | 424 | 381 |
|--------|-----|-----|-----|
| E2EPP | 0.5705±0.0082 | 0.5117±0.0132 | 0.4667±0.0099 |
| HAAP | 0.7126±0.0024 | 0.7010±0.0022 | 0.6594±0.0035 |
| RFGIAug | 0.7094±0.0021 | 0.6513±0.0026 | 0.6271±0.0031 |
| ReNAS | 0.6894±0.0019 | 0.6619±0.0033 | 0.6143±0.0027 |
| MLP | 0.6237±0.0018 | 0.5819±0.0023 | 0.5116±0.0011 |
| LSTM | 0.6837±0.0012 | 0.6349±0.0020 | 0.5874±0.0017 |
| *DARE* | **0.7576±0.0030** | **0.7311±0.0031** | **0.6814±0.0030** |

$$2 \exp \left( - \frac{2n^2t^2}{\sum_{i=1}^{n}(b_i - a_i)^2} \right) \leq \delta. \tag{11}$$

By transformation, we have:

$$n \geq \sqrt{\frac{\sum_{i=1}^{n}(b_i - a_i)^2}{2t^2} \ln \frac{2}{\delta}}. \tag{12}$$

Therefore, when we know the sample value and the probability upper bound $\delta$ we want, we can use the above formula to calculate the minimum sample size $n$ that meets the requirement of the probability upper bound.

In addition, the prerequisite for achieving an approximation of the sample distribution is to ensure that the sample is diverse, so it is necessary to combine the difference value $t$ with the sample diversity, which has:

$$t = \frac{1}{n(n-1)} \sum_{i=0}^{n} \sum_{j \neq i}^{n} (1 - \cos(X_i, X_j)) + \epsilon, \tag{13}$$

where $\cos(\cdot)$ is used to calculate the cosine similarity between samples and $\epsilon$ is a very small positive number to avoid the situation that the denominator is 0 in Equation 12.

## E DATASETS

This study mainly carries out experiments on search spaces, *i.e.*, NAS-Bench-101 (Ying et al., 2019), NAS-Bench-201 (Dong & Yang, 2020), and DARTS (Liu et al., 2018). The NAS-Bench-101 dataset contains 423K neural network architectures, each of which is trained, tested, and validated on CIFAR10 (Krizhevsky et al., 2009). The number of neural architectures in the NAS-Bench-201 is 15K, and each network architecture is trained on three image datasets, namely, CIFAR10, CIFAR100, and ImageNet16-120 (Russakovsky et al., 2015). DARTS is a method of differentiable architecture search that optimizes continuous parameters to find the best architecture. We tested our proposed method within the DARTS search space, which includes 7 nodes and 14 edges. In DARTS, each neural architecture is made up of two cells, a normal cell and a reduction cell. TransNAS-Bench-101 (Duan et al., 2021) is a multi-task neural architecture search (NAS) benchmark that provides performance data across seven tasks, including classification, regression, pixel-level prediction, and self-supervised learning. It features two types of search spaces: a macro-level space containing 3,256 architectures, and a cell-level space with 4,096 architectures. All architectures in both spaces have been trained, validated, and tested on the seven tasks.

NAS-Bench-101 and NAS-Bench-201 datasets are proposed to reduce expensive computational costs, and the users can quickly evaluate the performance of the neural architectures in the search space. In contrast, DARTS is used to validate the effectiveness of our approach in a larger search space, which does not contain performance metrics for the architecture. In addition, the neural architectures in all datasets can be conveniently transformed into graph structures, which facilitates the pre-processing of the data when we use the GCN model as a predictor.

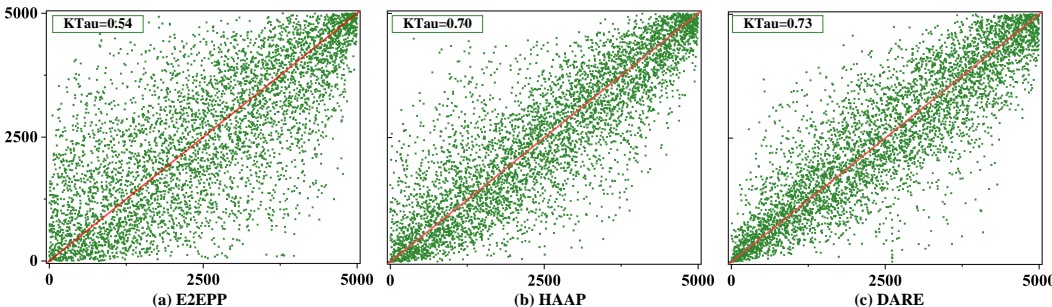

Figure 5: Qualitative comparison of *DARE* with E2EPP and HAAP algorithms. The x-axis represents the ground truth of the sample, and the y-axis represents the predicted value of the sample.

Table 7: Comparison results of *DARE* with the SOTA algorithms on NAS-Bench-201.

| Algorithms | $N_l$ | KTau | | | MSE | | |
|---|---|---|---|---|---|---|---|
| | | CIFAR10 | CIFAR100 | ImageNet16-120 | CIFAR10 | CIFAR100 | ImageNet16-120 |
| E2Epp | 150 | 0.6699±0.010 | 0.6620±0.080 | 0.6541±0.067 | 0.0013±0.0008 | 0.0024±0.0006 | 0.0022±0.0007 |
| HAAP | 150 | 0.7375±0.020 | 0.7184±0.009 | 0.7403±0.010 | **0.0005±0.0001** | 0.0015±0.0000 | 0.0010±0.0001 |
| RFGIAug | 150 | 0.7356±0.001 | — | — | — | — | — |
| RFGIAug | 78 | 0.7002±0.001 | — | — | — | — | — |
| MLP | 78 | 0.0974 | — | — | — | — | — |
| LSTM | 78 | 0.555 | — | — | — | — | — |
| HOP-2 | 78 | 0.5764 | — | — | — | — | — |
| *DARE* | 150 | **0.7854±0.005** | **0.7675±0.010** | **0.7906±0.010** | 0.0005±0.0002 | **0.0012±0.0003** | **0.0007±0.0001** |
| *DARE* | 78 | 0.7014±0.030 | 0.7198±0.020 | 0.7178±0.040 | 0.0008±0.0003 | 0.0021±0.0007 | 0.0013±0.0003 |

## F  RESULTS OF THE NEURAL PREDICTOR

In this section, we will give more comparison results on NAS-Bench-101 and NAS-Bench-201.

• **NAS-Bench-101.** For the NAS-Bench-101, we also perform a qualitative comparison and present the results in Fig. 5. The comparison algorithms include HAAP and E2EPP. We selected 424 samples as the training data and another 5000 samples randomly selected as the test data. In the figure, the x-axis represents the ground truth of the samples, the y-axis represents the predicted value of the samples, and the straight line $y = x$ is used as a reference, *i.e.* the more samples close to the straight line, the better the result. It can be seen from Fig. 5 that the effect of the *DARE* is the best, and the sample points are more concentrated. Additionally, we conducted more experimental results on NAS-Bench-101, as shown in Table 6, which similarly demonstrated the effectiveness of *DARE*.

• **NAS-Bench-201.** On NAS-Bench-201, we test all three image datasets (CIFAR10, CIFAR100, and ImageNet16-120). Table 7 shows the comparison results on the NAS-Bench-201. For the KTau, our algorithm achieves optimal results on all three image classification datasets, and the improvement is significant. On ImageNet16-120, for example, the *DARE* compared to E2EPP and HAAP improves by about 0.14 and 0.05 respectively. For the MSE, the *DARE* only shows a slightly larger bias than HAAP on CIFAR10, with the best performance for the rest of the image datasets. This indicates that our proposed algorithm can better predict the true performance of architectures than the other algorithms.

## G  RESULTS OF NAS

To further test the performance of the *DARE*, we apply the proposed algorithm to a real scenario to perform a search for the optimal network architecture. The same operation is also represented in (Liu et al., 2021; Xu et al., 2021), and the relevant experimental settings in this experiment are the same as in (Liu et al., 2021). Specifically, genetic algorithm (GA) (Sampson, 1976) is used as a search strategy to search the optimal architecture on three datasets (NAS-Bench-101, NAS-Bench-201, and DARTS). For the NAS-Bench-101 and NAS-Bench-201, the performance of the algorithm is

Table 8: Search results of **DARE** with SOTA algorithms on NAS-Bench-201.

| Algorithms | CIFAR10 | | CIFAR100 | | ImageNet16-120 | |
|---|---|---|---|---|---|---|
| | validation(%) | test(%) | validation(%) | test(%) | validation(%) | test(%) |
| RSPS | 84.16±1.69 | 87.66±1.69 | 59.00±4.60 | 58.33±4.34 | 31.56±3.28 | 31.14±3.88 |
| DATRS-V1 | 39.77±0.00 | 54.30±0.00 | 15.03±0.00 | 15.61±0.00 | 16.43±0.00 | 16.32±0.00 |
| DARTS-V2 | 39.77±0.00 | 54.30±0.00 | 15.03±0.00 | 15.61±0.00 | 16.43±0.00 | 16.32±0.00 |
| GDAS | 90.00±0.21 | 93.51±0.13 | 71.15±0.27 | 70.61±0.26 | 41.70±1.26 | 41.84±0.90 |
| ENAS | 39.77±0.00 | 54.30±0.00 | 15.03±0.00 | 15.61±0.00 | 16.43±0.00 | 16.32±0.00 |
| NPENAS | 91.08±0.11 | 91.52±0.16 | — | — | — | — |
| REA | 91.19±0.31 | 93.92±0.30 | 71.81±1.12 | 71.84±0.99 | 45.15±0.89 | 45.54±1.03 |
| RS | 90.03±0.36 | 93.70±0.36 | 70.93±1.09 | 71.04±1.07 | 44.45±1.10 | 44.57±1.25 |
| REINFORCE | 91.09±0.37 | 93.85±0.37 | 71.61±1.12 | 71.71±1.09 | 45.05±1.02 | 45.24±1.18 |
| BOHB | 90.82±0.53 | 93.61±0.52 | 70.74±1.29 | 70.85±1.28 | 44.26±1.36 | 44.42±1.49 |
| NASBOT | — | 93.64±0.23 | — | 71.38±0.82 | — | 45.88±0.37 |
| E2Epp | 90.61±0.89 | 93.39±0.75 | 71.08±2.00 | 71.11±1.93 | 44.36±1.85 | 44.87±1.43 |
| HAAP | 91.18±0.25 | 94.00±0.25 | 71.24±1.48 | 71.58±1.56 | 45.31±1.14 | 46.03±0.90 |
| ReNAS | 90.90±0.31 | 93.99±0.25 | 71.96±0.99 | 72.12±0.79 | 45.85±0.47 | 45.97±0.49 |
| *DARE* | **91.47±0.14** | **94.06±0.30** | **72.1±1.39** | **72.53±0.51** | **45.90±0.65** | **46.47±0.23** |
| ResNet | 90.83 | 93.97 | 70.42 | 70.86 | 44.53 | 43.63 |
| *optimal* | 91.61 | 94.37 | 73.49 | 73.51 | 46.77 | 47.31 |

evaluated by calculating the ranking of the selected optimal architecture in the whole search space. For the DARTS, we make the following settings. Since there are no corresponding metrics for the architectures in DARTS, we first trained the predictor by collecting 100 (or 60) architectures (tested on CIFAR10 to get the accuracy) using the AL strategy and then re-trained the optimal architectures obtained from the search on CIFAR10. To further validate the effectiveness of the algorithm, we also transferred the optimal architectures to the ImageNet dataset for testing. In addition, we set the maximum number of iterations to 20 for the GA and the population size to 100. The probabilities of crossover and mutation are 0.9 and 0.2, respectively. The experiment is repeated 20 times, and the Top-10 architectures are selected.

• **NAS-Bench-201.** Table 8 shows the search results on NAS-Bench-201, where we compare both the validation and test sets of three image datasets. The algorithms involved in the comparison can be divided into two categories, *i.e.*, algorithms provided in the seminal paper and algorithms based on predictors, which are listed in Table 8. In addition, the *optimal* represents the optimal architecture in the search space. Experimental settings are followed (Dong & Yang, 2020)

As shown in Table 8, **DARE** achieves optimal results on all three datasets. Compared to the *optimal*, the neural architectures obtained by **DARE** are all close or even equal in terms of accuracy. For example, on the CIFAR10 and CIFAR100 validation sets, **DARE** can search for the network architecture with the best performance, while on the CIFAR10 test set, the accuracy of the architecture searched by **DARE** differs from that of *optimal* by only 0.1%.

# H   ABLATION STUDY

The ablation study on NAS-Bench-201 (NB102) further validates the effectiveness of each component in our framework (**DARE**). As shown in Table 9, removing either the first-stage (w/o FS) or the second-stage (w/o SS) sampling strategy leads to a noticeable decline in model performance. The removal of the entire two-stage max-min sampling strategy (w/o TMMS) results in a sharp performance drop, highlighting the crucial role of TMMS in ensuring sample diversity and representativeness. In contrast, the impact of removing the key-point guided adaptive sampling strategy (w/o KGA) is more pronounced under the non-uniform distribution, demonstrating that KGA can effectively adapt to data imbalances by adaptively adjusting the sampling region to enhance performance in complex data environments.

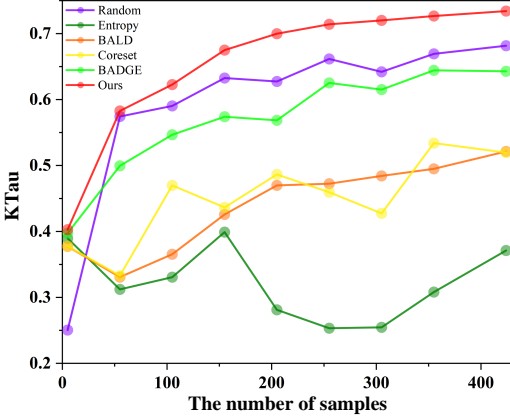

Figure 6: Performance comparison curves of our proposed method with other sampling strategies on NAS-Bench-101. All strategies select 424 samples. The x-axis indicates the number of samples selected and the y-axis indicates the corresponding KTau.

Table 9: Ablation Study on NB201.

| Distribution | Uni. | Non-Uni. |
|---|---|---|
| w/o FS | 0.7172 | 0.6634 |
| w/o SS | 0.7059 | 0.6791 |
| w/o TMMS | 0.5743 | 0.6118 |
| w/o KGA | 0.7258 | 0.7029 |
| *DARE* | 0.7714 | 0.7854 |

Table 10: performance on different samples.

| Predictors | selected samples | random samples |
|---|---|---|
| LR | **0.3969** | 0.35042 |
| RF | **0.5674** | 0.51668 |
| AdaBoost | **0.3917** | 0.33342 |
| Bagging | **0.4918** | 0.46574 |
| ExtraTree | **0.3541** | 0.31348 |

## I   OTHER EXPERIMENTS

**Sampling Strategies.** To verify the effectiveness of the proposed strategy, we compare it with other AL methods, including:

- **Random**: A baseline for the random selection of unlabeled samples.

- **Entropy** (Wang & Shang, 2014): A strategy of selecting samples with maximum entropy, which means the more information the sample contains.

- **BALD** (Gal et al., 2017): A strategy for calculating the uncertainty of unlabeled samples using the Bayesian method.

- **CoreSet** (Sener & Savarese, 2018): A strategy for selecting the most representative of the unlabeled samples and ensuring diversity of the labelled data.

- **BADGE** (Ash et al., 2020): A strategy that combines Bayesian and embedding to guarantee the uncertainty of unlabeled samples while considering the similarity of embedding.

The different methods select 424 samples on NAS-Bench-101. We record the change in KTau during the acquisition process. As shown in Fig. 6, the samples obtained by our strategy resulted in a leading performance of the predictor from the beginning to the end. In addition, our strategy improved by at least 0.05 in KTau compared to other strategies, while compared with Entropy, it has improved by more than 0.3. This demonstrates the effectiveness of our proposed strategy for performance predictors. However, there is an oddity in this experiment, *i.e.*, the Random method works better than other strategies. This is due to a data distribution problem with NAS-Bench-101, which we will detail in the **In-depth Analysis**.

**Neural Predictors.** To verify the generalizability of our proposed strategy, we have tested the selected samples using several predictors, including Linear Regression (LR), Random Forest (RF) (Breiman, 2001), AdaBoost (Freund & Schapire, 1997), Bagging (Breiman, 1996), and ExtraTree (Geurts et al.,

2006). In addition, we use randomly selected samples as controls. As shown in Table 10, "selected sample" denotes the samples selected by our strategy and "random samples" denotes the randomly selected samples. The sample sizes are both 424. In the table, all predictors perform better on samples obtained by our strategy than on samples obtained at random, which also proves the generalizability of our proposed algorithm.

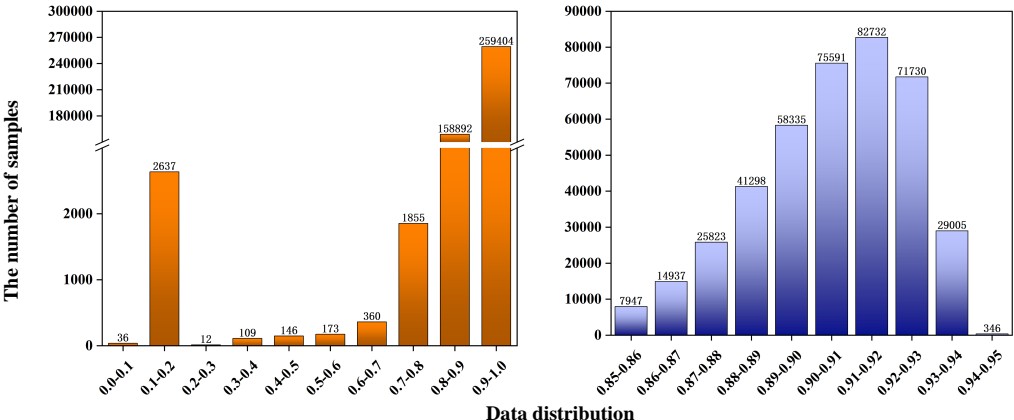

Figure 7: Data distribution based on test accuracy in NAS-Bench-101. The x-axis coordinate indicates the accuracy range, *e.g.*, "0.0-0.1" means that the test accuracy of the sample (network architecture) is between [0.0, 0.1], and the y-axis coordinate indicates the number of samples in the corresponding accuracy range. The left shows the distribution of all samples in the NAS-Bench-101 and the right shows the distribution of samples with test accuracy between [0.85, 0.95].

**In-depth Analysis.** In this section, we analyze the sample distribution in the NAS-Bench-101 and explain the question raised in **Sampling strategies** why Random would be more effective than other sampling strategies. Initially, we analyze the data distribution in the NAS-Bench-101. In Fig. 7, we plot the distribution of the data based on test accuracy, where the x and y axes indicate the accuracy range and the corresponding number of samples, respectively. As can be visualised in Fig. 7(left), although the range of accuracy is [0.0, 1.0], the number of samples with an accuracy greater than 0.8 accounts for nearly 99% of the total sample, which is the non-uniform distribution scenario. After further analysis, we find that the samples' accuracy is mainly concentrated between [0.85, 0.95], and therefore a more refined distribution is plotted in Fig. 7(right). Based on the above analysis, we can obtain that to train an NP with excellent performance, the training samples should be selected in an accuracy range of [0.85, 0.95]. Thus, as depicted in Fig. 8, the data distributions of the various sampling strategies are displayed. In the Fig. 8, the histograms show the number of samples, The red line and the orange line are both proportional curves (see the legend in Fig. 8 for details). Combining Fig. 8 with Fig. 6, it is clear that the performance of the NP is related to two factors, the number of samples selected in the accuracy range of [0.85,0.95] and the proportion of samples selected, both of which our proposed algorithm is optimal in. On the other hand, the main reason for the poor results of Entropy, BALD, CoreSet and BADGE is that too few samples are selected in the accuracy range [0.85, 0.95] and too many samples are selected outside the range, resulting in an imbalance in the training samples. In addition, as Random is free from human interference, the likelihood of selecting a sample is proportional to its frequency in the dataset. Thus, a large number of samples in the range [0.85, 0.95] increases the probability of selecting samples from this region, which coincidentally enhances the quality of the training samples and hence the predictor performance.

**Expansion Experiments on Transformer-based Structure.** To validate the effectiveness of our method in large-scale Transformer search spaces, we conducted experiments on the AutoFormer search space and compared it against baselines, including AutoFormer (Chen et al., 2021a), TF-NAS (Zhou et al., 2022), and AZ-NAS (Lee & Ham, 2024) (Table 11). The results clearly demonstrate *DARE*'s strong competitiveness. In the Small model comparison, *DARE* achieved the highest classification accuracy (83.1%) with the fewest parameters (22.7M). Similarly, in the larger Base model category, *DARE* achieved the highest accuracy on par with the baseline (82.4%) while maintaining the best parameter efficiency (53.8M). These results strongly prove that our method can

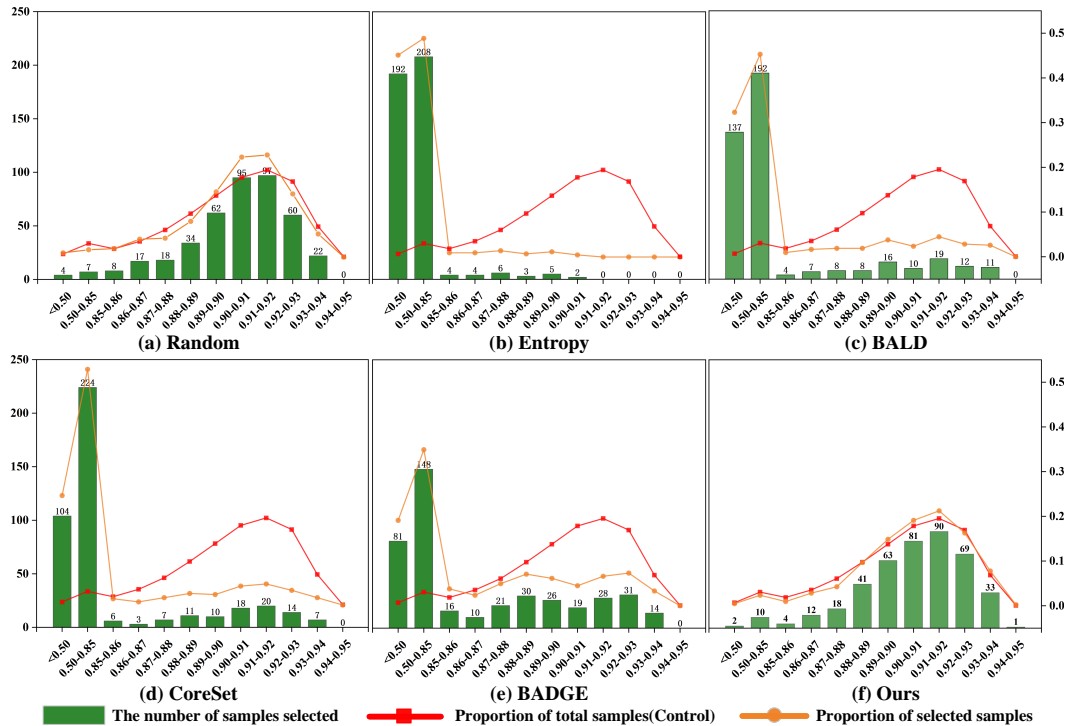

Figure 8: Distribution of samples selected by different sampling strategies. The x-axis coordinate indicates the accuracy range, the y-axis coordinate (left) indicates the number of samples selected by the sampling strategy in different accuracy ranges, and the y-axis coordinate (right) indicates the proportion of samples to the total samples. Specifically, the histogram is the number of samples. The red line refers to the proportion of the number of samples in the corresponding accuracy range to the total number of samples in the NAS-Bench-101. The orange line refers to the proportion of the number of samples selected by the sampling strategy in the corresponding accuracy range to the total number of samples selected (the total number is 424).

Table 11: Performance on Transformer-base Structure.

| Method | Small (Acc. (%) / Params) | Base (Acc. (%) / Params) |
|---|---|---|
| AutoFormer | 81.7 / 22.9M | 82.4 / 54.4M |
| TF-NAS | 81.9 / 23.0M | 82.2 / 56.5M |
| AZ-NAS | 82.2 / 23.8M | 82.3 / 54.1M |
| *DARE* | **83.1 / 22.7M** | **82.4 / 53.8M** |

be successfully extended to complex Transformer architecture search tasks, consistently discovering superior model architectures that balance both high accuracy and efficiency.

**Expansion Experiments on Various Tasks.** Table 12 presents the search results of *DARE* on the TransNAS-Bench-101 benchmark. We conducted experiments across all seven tasks, and the results show that *DARE* achieves the best performance on all tasks except for the Layout task, where it is slightly outperformed by REA. Notably, on several tasks across different search spaces, *DARE* even matches the *optimal* architecture. These results demonstrate that our method is effective not only for classification, but also for a wide range of non-classification tasks.

Table 12: Search results on TransNAS-Bench-101. **Bold** is the optimal value.

| Space | Methods | Accuracy (%) | | | L2 Loss ($\times 10^{-2}$) | mIoU (%) | SSIM ($\times 10^{-2}$) | |
|---|---|---|---|---|---|---|---|---|
| | | Scene | Object | Jigsaw | Layout | Segment. | Normal | Autoenco. |
| Micro | REA | 54.63 | 44.92 | 94.81 | **-62.06** | 94.55 | 57.10 | 56.23 |
| | RS | 54.56 | 44.76 | 94.63 | -62.22 | 94.53 | 56.83 | 55.55 |
| | HNAS | 54.29 | 44.08 | 94.56 | -64.83 | 94.57 | 56.88 | 48.66 |
| | RoBoT | 54.87 | 45.59 | 94.76 | -62.12 | 94.58 | 57.44 | 55.30 |
| | *DARE* | **54.91** | **45.59** | **95.01** | -62.34 | **94.61** | **58.03** | **56.33** |
| | *Optimal* | 54.94 | 45.59 | 95.37 | -60.10 | 94.61 | 58.73 | 57.72 |
| Macro | REA | 56.69 | 46.74 | 96.78 | -59.99 | 94.80 | 60.81 | 71.38 |
| | RS | 56.53 | 46.68 | 96.74 | -60.27 | 94.78 | 60.72 | 70.05 |
| | HNAS | 55.03 | 45.00 | 96.28 | -61.40 | 94.79 | 59.27 | 57.59 |
| | RoBoT | 57.41 | 46.87 | 96.89 | -58.44 | 94.83 | 61.66 | 73.51 |
| | *DARE* | **57.41** | **47.03** | **97.02** | **-58.22** | **94.86** | **62.17** | **73.89** |
| | *Optimal* | 57.41 | 47.42 | 97.02 | -58.22 | 94.86 | 64.35 | 74.88 |

