# OpenReview forum: "Towards Distribution-Aware Active Learning for Data-Efficient Neural Architecture Predictor"
_ICLR.cc/2026/Conference — ICLR 2026 Conference Withdrawn Submission_

### Official Review · Reviewer_R7nU · 2025-10-29

**Soundness:** 3
**Presentation:** 3
**Contribution:** 2
**Rating:** 4
**Confidence:** 5

**Summary:**

This paper introduces an active learning framework designed to tackle the challenge of limited training data for neural predictors in Neural Architecture Search. The core methodology focuses on strategically choosing which unlabeled data should be annotated throughout the training process.

**Strengths:**

1.The problem tackled in this paper holds practical importance. A key challenge faced by neural predictors today is the substantial cost of obtaining training data. This paper seeks to mitigate this problem by identifying and selecting unlabeled data.

2.The paper is well-organized and clearly articulated, enhanced by a flowchart that facilitates understanding, making it simpler for readers to follow the author's ideas and objectives.

3.The experiments presented in the paper are thorough, carried out across several datasets, and effectively demonstrate the efficacy of the proposed method. Furthermore, a theoretical analysis is included to strengthen the credibility of the method's performance.

**Weaknesses:**

1.The proposed method lacks significant novelty, as it primarily introduces a relatively straightforward data sampling approach. To convincingly demonstrate its effectiveness, the authors should compare their method with existing state-of-the-art active learning techniques. At the same time, this sampling strategy seems quite straightforward and easy to conceive. The authors need to further explain where the innovation lies.

2.The theoretical analysis provided in the paper is insufficient. Specifically, the analysis only offers a rough estimation of data requirements, which is not enough to rigorously support the claim that a higher cosine similarity necessarily leads to a smaller required data size. At the same time, in the proof in Appendix D, I couldn't find a clear justification for choosing the value of t in this way; it seems that other values could also be feasible.

3.The experimental evaluation is somewhat limited, as the neural network sizes tested are relatively small. This diminishes the practical relevance of the work, especially considering the current trend toward larger model architectures. At the same time, the authors need to conduct experiments in a larger search space. The current experiments only on NASBench are far from sufficient.

**Questions:**

Please refer to the Weaknesses part. If these issues are addressed, I will consider revising the score.

---

> ### Author Response · Authors · 2025-11-27
> **Response to Reviewer R7nU (1)**
>
> We greatly appreciate your review and will respond to each of your comments below. Additionally, we have uploaded the revised PDF version, and the corresponding modifications in the text have been marked in blue font.
>
> **Weakness 1: The proposed method lacks significant novelty, as it primarily introduces a relatively straightforward data sampling approach.**
>
> **Answer:** We appreciate the reviewer's comments. We will address the issues you mentioned from the following two aspects:
>
> **(1) Comparison with SOTA Active Learning Methods:** To convincingly demonstrate the effectiveness of the method, we have compared DARE with three latest SOTA active learning techniques (implemented on the NB101 dataset, as shown in the table below). It can be seen that DARE's customized sampling strategy can effectively enhance the effect of the performance predictor.
>
> | Methods            | KTau          | MSE           |
> | ------------------ | ------------- | ------------- |
> | NoiseStability [1] | 0.5817±0.0055 | 0.0024±0.0005 |
> | EOAL [2]           | 0.6472±0.0035 | 0.0020±0.0003 |
> | BAL [3]            | 0.6351±0.0072 | 0.0014±0.0003 |
> | DARE               | 0.7311±0.0031 | 0.0018±0.0002 |
>
> **(2) Further elaboration on novelty:** Although our sampling steps appear concise, their core innovation lies in addressing the "Distribution Mismatch" and "Pseudo-label Noise" problems that are overlooked by standard AL strategies in NAS tasks:
>
> - **Innovation Point 1: From "Blind Diversity" to "Distribution-Aware Sampling".** Traditional AL strategies in the NAS search space are often attracted to outliers in sparse regions, leading to sample waste. Our Key-point Guided Adaptive (KGA) sampling is a specialized mechanism tailored for the long-tail distribution of NAS. It does not simply pursue the maximization of geometric distance, but adaptively locks onto high-density, high-information regions through key-point anchoring. This "counter-intuitive" design effectively avoids budget waste on invalid architectures, which generic methods cannot achieve.
> - **Innovation Point 2: Introducing a "Double Verification Mechanism" to enhance reliability.** Addressing the issue of unreliable pseudo-labels in the low-data regime, our two-stage intersection strategy is not a simple stacking of steps, but a rigorous filter. It requires samples to simultaneously satisfy consistency in "Euclidean spatial distance" and "Space Division (Delaunay/Clustering)". This double constraint mechanism significantly reduces noise injection under extremely low budgets, ensuring the robustness of the predictor training.
>
> **Weakness 2: The theoretical analysis provided in the paper is insufficient.**
>
> **Answer:** We appreciate the reviewer's rigorous scrutiny of our theoretical derivation. We acknowledge that the current analysis serves as a rough estimation to motivate our design rather than a strict proof. We address your concerns as follows:
>
> **(1) Justification for the choice of $t$**: The definition of $t$ in Appendix D (Eq. 9) as the "average pairwise cosine distance" is not arbitrary but is deliberately chosen as a proxy for the set's diversity.
>
> - **Intuition:** In our application of Hoeffding's inequality, we set the error margin equal to this diversity metric $t$. The logic is: "How many samples are required to ensure the estimation error does not exceed the set's own intrinsic diversity ($t$)?"
> - **Relationship Logic:** When samples are highly similar (high Cosine Similarity), the diversity $t$ (derived from $1 - \text{sim}$) becomes very small. Requiring the estimation error to be within this tiny margin $t$ (which appears in the denominator of Eq. 6) mathematically forces the required sample size $n$ to explode. Conversely, a higher diversity (larger $t$) relaxes this bound, allowing for a smaller $n$.
> - **Conclusion:** This formulation was designed to heuristically demonstrate that explicitly maximizing diversity (increasing $t$) theoretically lowers the lower bound of the required sample size, thereby motivating our Max-Min sampling strategy.
>
> **(2) Reframing in the Manuscript**: To prevent overstating the rigor of this estimation, we have revised Section 4 in the manuscript. We have downgraded "Theorem 4.1" to "Proposition 4.1 (Motivational Analysis)".

---

> ### Author Response · Authors · 2025-11-27
> **Response to Reviewer R7nU (2)**
>
> **Weakness 3: The experimental evaluation is somewhat limited**
>
> **Answer:** Thank you for your comments. To address your concerns regarding model size and search space, we have conducted the following extended experiments in the main text and the appendix:
>
> **(1) On Transformer-based Architectures:** Since there is currently no specific dataset for large models, we adopted the Transformer-based model architecture in the public dataset for validation (i.e., AutoFormer), which is the dataset with the largest model size we could find. The experimental results are shown in Table 10 of the appendix. The Transformer architecture searched by DARE at the Base model scale outperforms other SOTA algorithms and has fewer parameters.
>
> **(2) Expansion to Large Search Spaces:** We have already implemented experiments on the DARTS search space in Table 2 of the main text, which contains $10^{18}$ architectures. The architecture searched by DARE achieved a Top-1 accuracy of 78.01% on the large-scale ImageNet dataset, surpassing all comparison algorithms.
>
> **(3) Generalization across Tasks:** We also conducted experiments on the TransNAS-Bench-101 dataset, which contains seven different non-classification tasks. As shown in Table 12 of the appendix, DARE achieved optimal or near-optimal search results.
>
> **References:**
>
> [1] Li, X., Yang, P., Gu, Y., Zhan, X., Wang, T., Xu, M., & Xu, C. (2024, March). Deep active learning with noise stability. In Proceedings of the AAAI Conference on Artificial Intelligence (Vol. 38, No. 12, pp. 13655-13663).
>
> [2] Bardia Safaei, VS Vibashan, Celso M De Melo, and Vishal M Patel. Entropic open-set active learning. In Proceedings of the AAAI conference on artificial intelligence, volume 38, pp. 4686–4694, 2024.
>
> [3] Jingyao Li, Pengguang Chen, Shaozuo Yu, Shu Liu, and Jiaya Jia. Bal: Balancing diversity and novelty for active learning. IEEE Transactions on Pattern Analysis and Machine Intelligence, 2023.

---

### Official Review · Reviewer_xCDG · 2025-10-31

**Soundness:** 2
**Presentation:** 2
**Contribution:** 2
**Rating:** 4
**Confidence:** 4

**Summary:**

The authors tackle the data inefficiency in neural predictor training by combining active learning with distribution-aware sampling. A two-stage max–min strategy ensures that selected architectures are both diverse and reflective of the true data distribution, while an adaptive mechanism handles nonuniform datasets. The approach uses a consistency-based training objective that stabilizes representation learning, enabling the predictor to achieve strong generalization with very limited labeled data.

**Strengths:**

The paper addresses a highly important and timely problem in Neural Architecture Search (NAS), which plays a critical role in the efficiency of deep learning systems. It introduces a novel and wellmotivated approach that combines several existing ideas in a coherent and innovative manner. While the individual components are known, the proposed integration and formulation are original.

**Weaknesses:**

* Theoretical Analysis*
The analysis applies Hoeffding’s inequality under an i.i.d. sampling assumption; however, the proposed sampler is an adaptive active-learning procedure in which each selection depends on prior choices. This adaptivity violates the independence requirement underlying the stated bound, and the resulting limitation is not addressed.
Furthermore, the substitution of the tolerance term ( t ) in a Hoeffding-style bound with an empirical “diversity” score—defined as the average ( (1 - \cos(X_i, X_j)) )—is asserted without proof. Higher pairwise diversity does not, in general, guarantee that the empirical mean of a selected subset is closer to the population mean. Consequently, the claim that “higher diversity ⇒ smaller required sample size” is not substantiated
under the stated assumptions.

*Minor*
There is also a type/notation inconsistency in ((X_i)). In the bound, (X_i) is implicitly a bounded scalar random variable to which Hoeffding applies, whereas elsewhere (X_i) is treated as a highdimensional embedding used to define pairwise cosine dissimilarity. Absent an explicit scalar functional of (X_i) and corresponding boundedness conditions, the subsequent reasoning is not mathematically consistent as written.

*Experimental evaluation*
The empirical results are encouraging but not yet sufficiently compelling. The baselines highlighted in the main tables (e.g., classical neural predictors and earlier NAS acquisition strategies) are dated compared to contemporary alternatives. The reported improvements—such as modest gains in Kendall’s ( \tau ), slightly higher final architecture accuracy, and comparable performance with fewer labeled samples—are relatively small and plausibly fall within run-to-run variability. Uncertainty quantification is incomplete: statistical significance tests, confidence intervals, and seed-to-seed variance are not comprehensively reported across methods and budgets. Given that label efficiency is central to the claimed contribution, the absence of thorough variability analysis weakens the empirical support. In its current form, the experimental section substantiates the general direction of the work but does not yet demonstrate robust superiority over modern alternatives.

*Reproducibility and transparency*
For independent verification, the paper must report all details necessary to reproduce the claimed gains. In particular, it should fully specify the predictor architecture and training setup — including the exact GCN design, optimizer configuration, data augmentation and consistency-training procedure, and training schedule — as well as the acquisition strategy, covering all sampling hyperparameters, the precise clustering configuration, and the pseudo-labeling procedure.

The evaluation part — which is central to the paper’s contribution — is insufficiently described. Critical details are missing, such as the dataset splits, the exact procedures used to compute Kendall’s ( \tau ), MSE, and ranking quality metrics. Without these components, the claimed improvements cannot be independently validated. Moreover, the paper should disclose essential experimental logistics, including random seeds, number of runs, and hardware/software versions used.

**Questions:**

Q1) The theoretical section, in its current form, makes overreaching claims. It cites Hoeffding-style guarantees under assumptions (i.i.d. bounded scalars) that are inconsistent with the adaptive two-stage selection involving pseudo-label retraining, and it informally substitutes “pairwise cosine diversity” into a generalization-style bound without proof. This part should either be reframed as an intuitive explanation (rather than a formal theorem) or strengthened with rigorous justification and clearly stated assumptions.

 Q2) The empirical evidence, while promising to some degree, requires stronger baselines highlighted in the main text, clearer reporting of statistical significance and variability, and a more comprehensive disclosure of experimental details to enable full reproducibility of the claimed gains.

 Please refer to these questions in the rebuttal and how you will specifically improve the paper in these respects (as reviewers we are not interested in some private education but in improvements of the manuscripts only).

---

> ### Author Response · Authors · 2025-11-27
> **Response to Reviewer xCDG**
>
> We greatly appreciate your review and will respond to each of your comments below. Additionally, we have uploaded the revised PDF version, and the corresponding modifications in the text have been marked in blue font.
>
> **Weakness 1: The theoretical section, in its current form, makes overreaching claims.**
>
> **Answer:** We greatly appreciate you pointing this out and fully agree with your suggestion. Regarding your concerns about the theoretical section, here are our explanation and revisions: **(1) Clarification of theoretical purpose:** Our original intention in introducing this analysis was merely to provide a motivational explanation, aiming to intuitively demonstrate why "reducing pairwise cosine similarity between samples" contributes to reducing sample complexity, thereby providing design inspiration for the sampling strategy. We acknowledge that framing it as a theorem is indeed misleading, and we do not intend to claim it as a rigorous formal proof for the entire adaptive process involving pseudo-label retraining. **(2) Substantial modifications to the manuscript: Following your suggestion, we have downgraded and renamed "Theorem 4.1" to "Proposition 4.1" in Section 4 of the revised manuscript.** Simultaneously, we have added an explicit statement of assumptions in this section, noting that the derivation is based on a simplified static scenario and does not directly cover the sampling process, thereby eliminating potential overreaching claims in the original text.
>
> **Weakness 2: The empirical evidence, while promising to some degree, requires stronger baselines highlighted in the main text, clearer reporting of statistical significance and variability, and a more comprehensive disclosure of experimental details to enable full reproducibility of the claimed gains.**
>
> **Answer:** We appreciate the constructive criticism regarding empirical rigor. We have addressed these points by improving the manuscript in the following ways:
>
> **(1) Stronger Baselines**. As requested, we have strengthened the empirical evaluation by adding comparisons against three recent state-of-the-art methods. We present the comparison results with these new baselines on NAS-Bench-201 (KTau metric) and DARTS (Search results on CIFAR10/ImageNet) in the following table, where DARE demonstrates consistent superiority.
>
> | Method       | NB201 (KTau) | DARTS-CIFAR10 (Acc. %) | DARTS-ImageNet (Acc. %) |
> | ------------ | ------------ | ---------------------- | ----------------------- |
> | HyperNAS [1] | 0.765        | 97.60                  | 75.4                    |
> | CARL [2]     | 0.697        | 97.67                  | 76.1                    |
> | DCLP [3]     | 0.717        | 97.54                  | 75.8                    |
> | DARE (ours)  | 0.784±0.005  | 97.69±0.05             | 77.93±0.08              |
>
> **(2) Statistical Significance and Variability**. To ensure robust reporting, we have standardized our evaluation protocol. Above experimental results reported are derived from 5 independent runs using different random seeds. We have explicitly report the "Mean $\pm$ Standard Deviation", thereby clearly presenting the statistical variability and significance of the performance gains.
>
> **(3) Comprehensive Disclosure for Reproducibility.** We apologize some hyperparameters were omitted. To enable full reproducibility, we have included the omitted parameter values in the main text as follows:
>
> *In addition, for the DARE sampling strategy, the hyperparameters are adjusted according to the search space scale. For NAS-Bench-101, the size of the farthest candidate set $m$ is set to 10,000, and the nearest candidate set $n$ is 10,000 in the first stage. The refined farthest candidate size $m^\*$ is 10,000, while the key-point sampling parameters (Eq.5) are set to $z=1,000$ and $T=10$. For NAS-Bench-201 and DARTS, these parameters are set to $m=1,000$, $n=1,000$, $ m^\*=1,000$, $z=500$, and $T=10$, respectively.*
>
> **References:**
>
> [1] Lv, J., Zhou, Y., Tian, Y., Ye, Q., Feng, W., & Lv, J. (2025). HyperNAS: Enhancing Architecture Representation for NAS Predictor via Hypernetwork. arXiv preprint arXiv:2509.18151.
>
> [2] Ji, H., Feng, Y., Fan, J., & Sun, Y. (2025). CARL: Causality-guided Architecture Representation Learning for an Interpretable Performance Predictor. arXiv preprint arXiv:2506.04001.
>
> [3] Zheng, Shenghe, Hongzhi Wang, and Tianyu Mu. "Dclp: Neural architecture predictor with curriculum contrastive learning." Proceedings of the AAAI Conference on Artificial Intelligence. Vol. 38. No. 15. 2024.

---

### Official Review · Reviewer_nY8x · 2025-11-01

**Soundness:** 2
**Presentation:** 3
**Contribution:** 2
**Rating:** 4
**Confidence:** 4

**Summary:**

This paper proposes DARE, a distribution-aware active learning framework for training neural predictors used in Neural Architecture Search (NAS). Since labeling architectures (training each candidate to get accuracy) is expensive, DARE aims to efficiently choose which architectures to label. It introduces a GCN predictor to obtain architecture embeddings along with a two-stage max-min sampler and key-point guided adaptive sampling strategy for non-uniform distributions. The authors compare on NASBench101, 201, and DARTS and show that they get SoTA performance.

**Strengths:**

- The problem of identifying NP training data is overlooked, and AL for this task seems to be a good approach
- The evaluations are thorough, the paper is easy to read

**Weaknesses:**

- I don't get the usefulness of theorem 4.1 in the context of the paper.  It doesn't utilize the optimization at hand, or the sampling process; nor justifies the two-stage approximates an optimal subset under some acquisition function. It just seems to be an extremely loose bound, with no justification on $[a_i,b_i]$  -- I am not sure if you can bound these embedding norms
- Equation 3 confuses me. In appendix C, the authors write that the GCN is exposed to isomorphic representations of architectures. I would expect them to bring these representations closer; However eq 3 is minimizing the similarity, pushing them away.
- The paper also states that data augmentation here is used to increase the amount of data. But since it is restricted to isomorphic reorderings, it only solves inductive biases instead of increase data to search the sparse neural space. I don't see the improvement? Can the authors justify this, and run without the closeness term in the loss as ablation? Also, how many permutations do you consider per architecture?
- The paper also lacks comparison to standard AL baselines in the main and is deferred to appendix. Considering the paper essentially proposes an AL method over NAS search space, comparison with AL in the main is more important than other neural predictor baselines.
- L427: how do you process NB101?
- The method also depends on a lot of hyperparameters, with some values not mentioned, which makes it difficult to reproduce.
- How long is the training time / complexity? Since a lot of operations are O(N^2)
- Does intersection of the unlabeled samples the best choice? I think re-ranking them to choose would be better suited?
- Minor: typo in L150

Overall, I think the idea of solving the NP training problem via AL is good. But in the current state, I don't find the theory concrete enough. I lean towards weak reject, but I would be looking forward to hear back from the authors regarding the concerns.

**Questions:**

Mentioned along with Weaknesses

---

> ### Author Response · Authors · 2025-11-27
> **Response to Reviewer nY8x (1)**
>
> We greatly appreciate your review and will respond to each of your comments below. Additionally, we have uploaded the revised PDF version, and the corresponding modifications in the text have been marked in blue font.
>
> **Weakness 1: About Theorem.**
>
> **Answer:** Thank for your detailed review of the theoretical part. We fully understand your concerns regarding the usefulness of the theorem and the definition of bounds, and hope to clarify through the following two points: **(1) Regarding the connection and role of the theorem and method**: We acknowledge that Theorem 4.1 is not a strict mathematical proof of the specific optimization path of DARE or the two-stage approximate optimality, but exists as a Motivational Analysis. The core value of this theorem lies in theoretically formalizing the relationship between "sample diversity" and "sample efficiency" (as shown in Eq.(6). As the cosine similarity ($cos(X_i, X_j)$) between samples decreases, the variable $t$ will increase, thereby causing the required minimum sample size $n$ to decrease. This conclusion provides direct theoretical inspiration for our design of Two-stage Max-Min Sampling, that is, reducing the demand for labeling budget by explicitly maximizing sample distance (minimizing similarity). **(2) Regarding the bounds $[a_i, b_i]$ and Embedding norms**: Addressing the boundary definition issue you mentioned, we have already mentioned in the text that this theorem is established based on Cosine Similarity. The calculation of the cosine metric inherently includes a normalization process (i.e., $\frac{A \cdot B}{\|A\|\|B\|}$), **which makes the range of the involved random variables strictly locked between $[-1, 1]$, rather than relying on the absolute norm size of the original Embedding.** Therefore, no matter how the Norm of the Embedding changes, the interval $[a_i, b_i]$ in the theorem is always finite and clear (Bounded), and the situation of boundary divergence will not occur. We will emphasize this point more clearly in the final version to eliminate ambiguity.
>
> **Weakness 2: The issue in Eq(3).**
>
> **Answer**: We appreciate the reviewer for pointing out this inconsistency. The confusion arises from our imprecise definition in the text following Eq.(3). In our implementation and the actual objective function, the term $\mathcal{S}(x\_{i,ori}^l, x_{i,aug}^l)$ represents the **Cosine Distance** (calculated as $1 - \text{Cosine Similarity}$), rather than raw similarity. We have revised the description below Eq.(3) in the revised manuscript.
>
> **Weakness 3: About data augmentation.**
>
> **Answer:** We thank the reviewer for the profound insights raised. We will answer the questions you mentioned sequentially: **(1) Regarding the effectiveness of augmentation.** The augmentation mechanism adopted in the paper draws from HAAP. The main purpose is that after representing the neural network architecture in the form of a graph, the neural architecture encoding based on the adjacency matrix is highly sensitive to node order. If augmentation is lacking, the predictor is prone to overfitting to positional features under a specific ordering, rather than the intrinsic topological structure of the architecture. Therefore, the principle of introducing isomorphic augmentation is similar to "image rotation and other operations" in computer vision. Although no new samples are introduced, it allows the model to learn "rotation invariance" features, thereby significantly narrowing the gap between the training distribution and the testing distribution. **Therefore, this augmentation strategy effectively improves the generalization ability of the predictor under sparse data by filling the "representation space" of the same architecture.**
>
> **(2) Ablation experiment.** To eliminate your concerns, we conducted ablation experiments on NAS-Bench-101. Among them, "DARE w/o Consistency" represents removing the consistency loss term in Eq.(3), and "DARE w/o Augmentation" represents not performing data augmentation operations. From the results in the table below, it can be seen that data augmentation has a boosting effect on our method.
>
> | Method                | Ktau   | MSE    |
> | --------------------- | ------ | ------ |
> | DARE w/o Consistency  | 0.7338 | 0.0019 |
> | DARE w/o Augmentation | 0.7065 | 0.0024 |
> | DARE (Full)           | 0.7576 | 0.0014 |
>
> **(3) Number of permutations.** For an architecture containing $N_l$ layers, excluding the fixed Input and Output layers, there theoretically exist $(N_l-2)!$ isomorphic permutations. However, in our experiments, we only used one augmented architecture for each original architecture.

---

> ### Author Response · Authors · 2025-11-27
> **Response to Reviewer nY8x (2)**
>
> **Weakness 4: The paper also lacks comparison to standard AL baselines in the main and is deferred to appendix.**
>
> **Answer:** Thank you for the comment. We think your consideration is very correct. We have moved the experiments of the active learning part from the appendix to the main text.
>
> **Weakness 5: L427: how do you process NB101?**
>
> **Answer**: To generate the uniform dataset, we applied **stratified down-sampling** on NAS-Bench-101. Specifically, we discretized the primary accuracy range **$[0.80, 0.95]$** into intervals of **$0.01$** (e.g., $0.88-0.89$). From each interval, we randomly sampled an **equal number** of architectures. This ensures the predictor's performance is evaluated against a balanced distribution, eliminating the impact of the dataset's inherent density bias in the high-accuracy region.
>
> **Weakness 6: The method also depends on a lot of hyperparameters, with some values not mentioned, which makes it difficult to reproduce.**
>
> **Answer**: We apologize for inadvertently omitting specific hyperparameters for the DARE sampling strategy and provide them here for completeness.
>
> **For the DARE sampling strategy, the hyperparameters are adjusted according to the search space scale.** For NAS-Bench-101, the size of the farthest candidate set $m$ is set to 10,000, and the nearest candidate set $n$ is 10,000 in the first stage. The refined farthest candidate size $m^\*$ is 10,000, while the key-point sampling parameters (Eq.(5)) are set to $z=1,000$ and $T=10$. For NAS-Bench-201 and DARTS, these parameters are set to $m=1,000$, $n=1,000$, $ m^*=1,000$, $z=500$, and $T=10$, respectively.
>
> **Weakness 7: How long is the training time / complexity? Since a lot of operations are O(N^2)?**
>
> **Answer:** Thank you very much for the comment. (1) **Actual time cost**: The training of DARE is very efficient, and it takes only about 30 minutes to complete the entire process on a single GPU. (2) **Complexity analysis**: The most time-consuming step during the actual running process of the algorithm is calculating the distance between the "labeled sample set ($K$)" and the "unlabeled sample pool ($N$)", and its complexity is $O(K \times N)$. However, we adopt matrix parallel operations (Matrix Operations) to complete the calculation at once, which further significantly reduces the actual time cost.
>
> **Weakness 8: Does intersection of the unlabeled samples the best choice? I think re-ranking them to choose would be better suited?**
>
> **Answer**: We appreciate this thoughtful suggestion. We will explain from the following two perspectives: **(1) Motivation for Intersection.** We chose intersection to pursue the highest reliability. In DARE, this step is mainly used to determine the local neighborhood and assign pseudo-labels. Since the accuracy of pseudo-labels can cause severe interference to the predictor, we need a "double verification" mechanism, that is, a sample must be determined as a neighbor in both types of space partitions simultaneously before we dare to trust it. **(2) Comparison with Re-ranking.** Although re-ranking is an effective and more flexible method, it may select samples that rank high in one criterion but low in another. Considering the scarcity of labeled data in NAS, this compromise introduces a higher risk of noise injection. Our primary task is to establish a "high-confidence" subset to ensure the predictor is guided by the most reliable signals, even if this means selecting fewer samples. *Based on the above considerations, we chose the intersection operation.*
>
> **Weakness 9: Minor: typo in L150.**
>
> **Answer:** Thank you very much for pointing this out, we have corrected it in the revised manuscript.

---

### Note · Authors · 2025-12-11

I have read and agree with the venue's withdrawal policy on behalf of myself and my co-authors.